# An interactive ImageJ plugin for semi-automated image denoising in electron microscopy

Joris Roels[1,2,7]*, Frank Vernaillen [3,4,7], Anna Kremer[1,4,5], Amanda Gonçalves[1,4,5], Jan Aelterman[6], Hiêp Q. Luong [6], Bart Goossens[6], Wilfried Philips[6], Saskia Lippens[1,4,5,8] & Yvan Saeys [1,2,8]

The recent advent of 3D in electron microscopy (EM) has allowed for detection of nanometer resolution structures. This has caused an explosion in dataset size, necessitating the development of automated workflows. Moreover, large 3D EM datasets typically require hours to days to be acquired and accelerated imaging typically results in noisy data. Advanced denoising techniques can alleviate this, but tend to be less accessible to the community due to low-level programming environments, complex parameter tuning or a computational bottleneck. We present DenoisEM: an interactive and GPU accelerated denoising plugin for ImageJ that ensures fast parameter tuning and processing through parallel computing. Experimental results show that DenoisEM is one order of magnitude faster than related software and can accelerate data acquisition by a factor of 4 without significantly affecting data quality. Lastly, we show that image denoising benefits visualization and (semi-)automated segmentation and analysis of ultrastructure in various volume EM datasets.

[1] VIB, Center for Inflammation Research, Technologiepark 71, B-9052 Ghent, Belgium. [2] Ghent University, Department of Applied Mathematics, Computer Science and Statistics, Krijgslaan 281-S9, B-9000 Ghent, Belgium. [3] VIB, Bioinformatics Core, Rijvisschestraat 126 3R, B-9052 Ghent, Belgium. [4] VIB, Bioimaging Core, Technologiepark 71, B-9052 Ghent, Belgium. [5] Ghent University, Department of Biomedical Molecular Biology, Technologiepark 71, B-9052 Ghent, Belgium. [6] Ghent University/IMEC, Department of Telecommunications and Information Processing, St-Pietersnieuwstraat 41, B-9000 Ghent, Belgium. [7] These authors contributed equally: Joris Roels, Frank Vernaillen. [8] These authors jointly supervised this work: Saskia Lippens, Yvan Saeys. *email: joris.roels@ugent.vib.be

The field of three-dimensional electron microscopy (3D EM) covers several technologies that unveil a sample at nanometer (nm) resolution. The classical setup typically involves serial sectioning and high-resolution imaging by transmission EM (TEM). Lots of progress has been made in this field by automating acquisition, which eventually enabled successful imaging of the complete *Drosophila melanogaster* brain at synaptic resolution[1]. The development of serial block face (SBF) scanning EM (SEM) techniques has made 3D EM more easily available for large-scale imaging of biological samples[2]. SBF-SEM repetitively acquires a 2D SEM image from the smoothened sample surface (or block face) and then removes the top of the sample with a diamond knife ultramicrotome[3,4], revealing the next sample surface to be imaged. Eventually, this results in a stack of 2D images that can be compiled to a high-resolution 3D volume image. A similar slice-and-view approach is used in focused ion beam (FIB) SEM, where the block face is removed by FIB milling. While both SBF-SEM and FIB-SEM have the potential to generate images at 3- to 5-nm lateral resolution, the FIB milling is more precise than the mechanical SBF-SEM slicing, resulting in a maximal axial resolution of 5 and 20 nm, respectively[2,5,6].

Over the past years, there has been a substantial increase in the use of these techniques in life science research[7–12]. The advantage of generating high-resolution 3D information, and also a comprehensive view of a complete cell or tissue, has invited the scientific community to apply these techniques for many different research questions. Recent ambitious research projects, such as imaging $10^7$ µm³ sections of Drosophila brain and mammalian neuronal tissue[12,13] at 8 nm³ isotropic resolution for connectomics research have taken volume EM imaging to a next level. Even considering the impressive tenfold speedup obtained by Xu et al.[13], it still requires 6 months and six FIB-SEM machines to section an entire Drosophila ventral nerve cord of $\sim 2.6 \times 10^7$ µm³ voxels. Note that the classical image acquisition setup with a single FIB-SEM machine, used by most other research facilities, would require more than 5 years. Consequently, this approach is limited in terms of scalability. A potential solution arises in the dwell-time acquisition parameter, i.e., the time that is used to "illuminate" one pixel. Shorter dwell times have two advantages: shorter total acquisition time and less risk to overexposure artefacts such as charging. However, the noise level increases as the dwell time decreases, which can introduce issues with regard to subsequent visualization, segmentation, and analysis of ultrastructure.

For the last few years, there has been great progress in computer vision research, particularly in image denoising, which aims to restore the true image signal from noisy data. State-of-the-art denoising methods are based on multiresolution shrinkage[14,15], nonlocal pixel averaging[15,16], Bayesian estimation[17,18], or convolutional neural networks[19]. Many of these methods have shown remarkable performance for 3D EM applications[20–25]. Even though most of these methods are available to the community, they are often impractical due to low-level programming environments, parameter sensitivity, and high computational demands. We believe that an interactive approach is required as the restored image data can only be validated by experts. Nevertheless, the existing interactive denoising frameworks[26–28] tend to rely on parameters that are difficult to interpret and/or are computationally too intensive for efficient tuning and scaling toward large-scale 3D data sets, such as the teravoxel size data sets generated by Xu et al.[13] Furthermore, the current state-of-the-art in image restoration is evolving fast, prompting the need for a framework that is easily extendible with new algorithms.

In this work, we propose an interactive and user-friendly framework called DenoisEM equipped with state-of-the-art image restoration algorithms, combined with intuitive parameter interpretation available through ImageJ[29]. The computational backend is accelerated via GPU-based massive parallel computing and a high-level programming language called Quasar[30]. We show that by using DenoisEM data acquisition times can be reduced by a factor of 4 without significantly affecting image quality. We also show that visualization and automated segmentation and analysis can be improved by using the denoising algorithms that are implemented in DenoisEM. Our plugin is publicly available at http://bioimagingcore.be/DenoisEM.

## Results

**Interactive semi-automated image restoration with DenoisEM.** Many solutions have been proposed for restoration of EM images[31]. However, practical solutions that allow for user feedback to apply state-of-the-art denoising on large 3D EM data sets generated by e.g., SEM or serial section TEM are not readily available. Optimal finetuning of parameters in denoising is crucial, and this requires expert intervention. Therefore, we wanted to offer a tool that is based on a human-in-the-loop approach. To tackle this challenge, we developed DenoisEM, a GPU accelerated denoising plugin with an interactive workflow that allows for efficient interaction and feedback by the expert. DenoisEM is a plugin for ImageJ[29], an open-source program that is extensively used in the microscopy community. The plugin allows for quick testing, comparison, and application of different denoising solutions, and can be used for any modality that generates 3D image data. The plugin workflow (see Fig. 1) consists of six steps: data loading, initialization, region-of-interest (ROI) selection, noise estimation, interactive parameter optimization, and final batch processing. Each step is automated as much as possible, and user interaction is only required in the selection of the ROI and parameter settings. DenoisEM is highly optimized and offers parameter tuning at low latency due to a GPU accelerated back-end engine called Quasar.

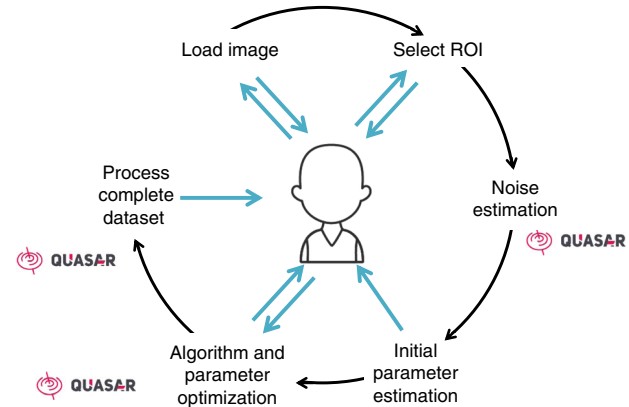

**Fig. 1 Graphical workflow of our proposed framework that includes a human in the loop.** An image is loaded and the computation backend is prepared. Next, the user selects a ROI that is representative for the complete data set. The noise level is automatically estimated to derive near optimal parameter initialization (see "Parameter estimation" section in Supplementary Figs. 16, 17). Next, the biological expert can optimize the parameter settings at a low latency visualization of the results according to their preferences (typically w.r.t. visualization and/or subsequent segmentation of specific objects). Once the optimal parameters for a specific algorithm are found, the complete data set is ready to be processed. The computationally intensive parts of the workflow are GPU accelerated and indicated with the Quasar logo[30].

**Intuitive and interactive user interface**. DenoisEM guides the user through the denoising process via a simple interactive user interface (see Fig. 2) in a step-by-step procedure:

- The user starts by opening a 2D or 3D image in ImageJ. During startup of the plugin, this image is assigned as the reference image for restoration and all computational resources (CPU and, if available, GPU) are prepared (Fig. 2a).
- Next, the user can select a particular (2D) region-of-interest (ROI) in the reference image that is representative for the data set and for the analysis that follows restoration (e.g., visualization or segmentation of specific structures). This can be intuitively performed using the available selection tools in ImageJ. At this point, the noise level is automatically estimated on the ROI, so that initial parameter settings of the available algorithms are at a favorable starting point for further finetuning in the next step. For more details regarding the noise and parameter estimation, we refer to the respective "Methods" sections and Supplementary Figs. 16, 17.
- The next step involves algorithm selection and interactive finetuning of the parameter settings. For high-quality image restoration this needs to be done carefully for each data set, because EM data can be highly variable due to different factors, like sample preparation procedures, acquisition circumstances, cell type, etc. Moreover, each denoising algorithm has its advantages and disadvantages and depending on the data and parameter settings the result can

vary significantly (see Supplementary Figs. 2–10). For example, Gaussian filters are very effective in noise removal at the cost of edge blurring (for large values of $\sigma$), while the nonlocal means filter is ideal for the data with many repetitive patterns, but computationally intensive (for large window sizes). Note that denoising inherently always boils down to a trade-off between noise reduction and sharpness. For use cases where sharpness is of high importance, we offer an optional panel that estimates the sharpness of the image[32].

A strong asset of DenoisEM is that different algorithms (listed and briefly discussed in "Methods"), including recently developed ones, can be found in one single tool and are practical to use due to GPU acceleration. Switching between algorithms is done by checking the corresponding box and different parameters can be set by sliders. Typically, the influence of different parameter settings is demonstrated at low latency, which facilitates the finetuning process. When visualization would lag, it is indicated by a red frame around the denoised ROI. Tooltips indicate the influence of the parameter on the result (e.g., noise reduction or sharpening). When the user proceeds to a different algorithm or to the next step, the previous parameters are cached, making it feasible to switch back if necessary.

- In the final step, the user can apply the desired denoising algorithm and its corresponding parameters on the complete image. A new image is generated and the user can then save

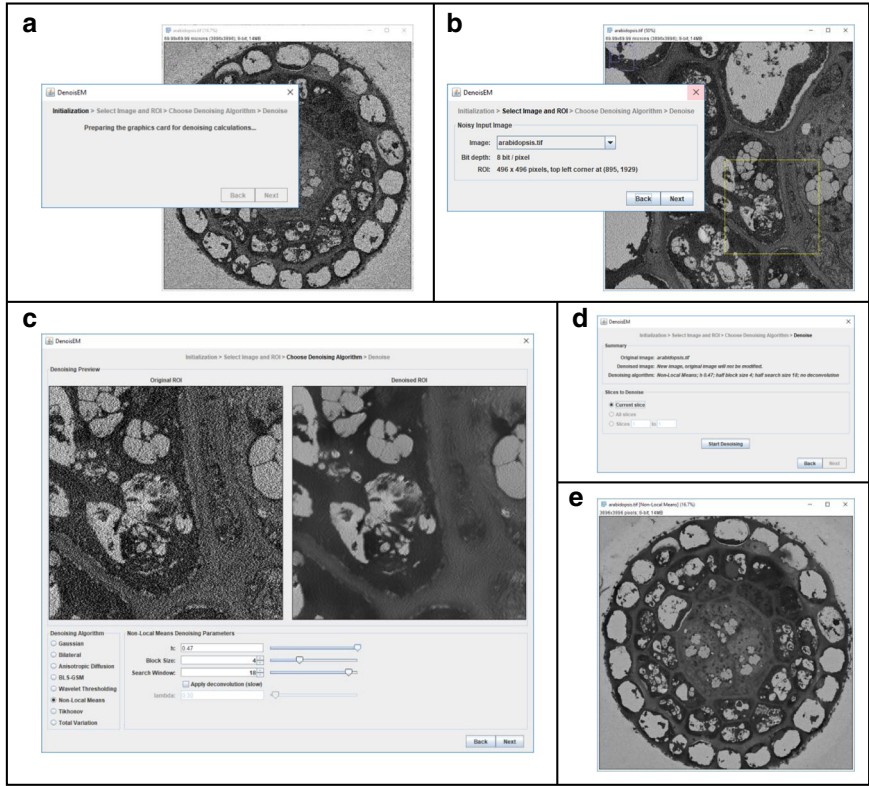

**Fig. 2 An overview of the user interface (UI) of the DenoisEM plugin for ImageJ.** It is structured as a wizard that guides the user through the denoising process in a sequence of steps. **a** In ImageJ, the user loads an image or image stack for denoising and starts the DenoisEM plugin. The UI wizard appears, and the computational backend for parallel processing on the CPU/GPU (Quasar) is initialized. **b** In the next step, the user chooses any of the open images for denoising and selects a ROI on which denoising will be previewed. **c** Next, the main panel in the plugin appears. At the top it shows side-by-side the noisy original as well as the denoised version of the selected ROI. In the bottom left corner, the user can select one of eight denoising algorithms. The bottom right has controls for specifying the algorithm parameters. Typically, if the algorithm or its parameters are changed, the denoised ROI at the top is updated virtually instantaneously. This allows the user to easily assess the effect of algorithms and parameter settings. **d** After optimal settings are chosen, the user is shown a short summary and (for image stacks) can select the image slices that need to be denoised. During denoising, the user is shown progress feedback. **e** When denoising is finished, a new image or image stack is created and displayed. The original image (stack) is left untouched.

the denoised image using ImageJ. We recommend to save the image in TIFF format in order to store the metadata that also contain the information on the denoising algorithm and parameters. These parameter values can then be applied later on to new data sets.

We provide eight different denoising algorithms in DenoisEM: Gaussian filtering, wavelet thresholding[24,33–37], anisotropic diffusion[38–40], bilateral filtering[41–43], Tikhonov denoising/deconvolution[44], total variation denoising[45], Bayesian least-squares Gaussian scale mixtures[14], and nonlocal means denoising/deconvolution[16,18,25,46]. For more details about these algorithms, we refer to the Methods section. Supplementary Figs. 2–10 show the effect of applying each of these denoising algorithms under several different parameters on the same noisy image patch. The plugin user manual, available through Supplementary Note 3, describes the different algorithms and the associated parameters from a user perspective.

Because the DenoisEM plugin allows to select any image that is open in the program, it allows for alternating with other applications in ImageJ. After each step in the workflow, it is possible to go back to previous steps and at no point the original image is overwritten.

**Improved visualization of 3D EM ultrastructure.** We used DenoisEM on SBF-SEM images of an Arabidopsis thaliana root tip. The en bloc stained sample was prepared as described[47,48], and a SBF-SEM data set was recorded with a Zeiss Merlin with 3View. The original image contained a significant amount of noise and we used the DenoisEM plugin to test different denoising solutions, under visual assessment by a biology expert. Denoising was done by applying Tikhonov deconvolution ($\lambda = 1.5$, $\sigma = 0.31$, and $N = 86$ iterations), and Fig. 3 shows the result of the denoising on four different ROIs from the data sets, and for each ROI both an XY and a YZ orthogonal view is shown. In this particular case, mild denoising was applied to avoid the loss of structural details in the image. Nevertheless, there was an effect on image quality and improved recognition of certain subcellular structures, e.g., the nuclear membrane (Fig. 3b, ROI 1) or

endoplasmic reticulum (Fig. 3b, ROI 3). Note that the image quality has also improved in the axial direction, even though the plugin performs lateral slice-by-slice denoising. We found that the noise level was decreased by two orders of magnitude (Fig. 3c) using a median absolute deviation noise level estimator.

In a next example, we used SBF-SEM images of mouse heart tissue, prepared and imaged as described by Vanslembrouck et al.[49] Figure 4a shows an unprocessed 2D image of heart smooth muscle cells. The sarcomeres can be recognized with the A-bands as darker zones and the I-bands as brighter zones. We applied the anisotropic diffusion algorithm (step size $\eta = 0.07$, $N = 6$ iterations and a diffusion factor $\kappa = 0.18$) on this data using DenoisEM. The effect of denoising is shown in Fig. 4b, while Fig. 4e, f shows the corresponding intensity histogram before and after denoising. Denoising is beneficial for the interpretation of the sarcomere organization, since the A- and I-bands can be better distinguished after noise removal. This is illustrated in Fig. 4c, d, where intensity thresholding is used to generate a magenta and green mask for the A- and I-bands, respectively.

**Increased throughput of 3D EM imaging.** Noise levels inversely correlate with the acquisition dwell time, meaning that long dwell times will always give a better result with respect to noise levels in an image. However, for SBF-SEM, long dwell times have the overall downside that the generation of a single data set can result in multiple imaging days. In addition, this also leads to electron charging artifacts and difficult slicing of the specimen. Ideally, shorter dwell times should be used, without the trade-off of too much noise in the image. Therefore, we acquired an image at 1 μs dwell time and used DenoisEM to generate a denoised 1-μs image and compared this to images acquired at longer dwell times of 2 μs and 4 μs (Fig. 5). The general image quality of the fast denoised image is improved, and for recognition of structures this can correspond to the image that was acquired at 4 μs. This means that the use of fast denoised images, instead of slower imaging, has an impact on the general acquisition time. If data are generated at 10-nm pixels, with 100 nm slicing of a block of $100 \times 100 \times 500$ μm$^3$, this means that 500x images are acquired and

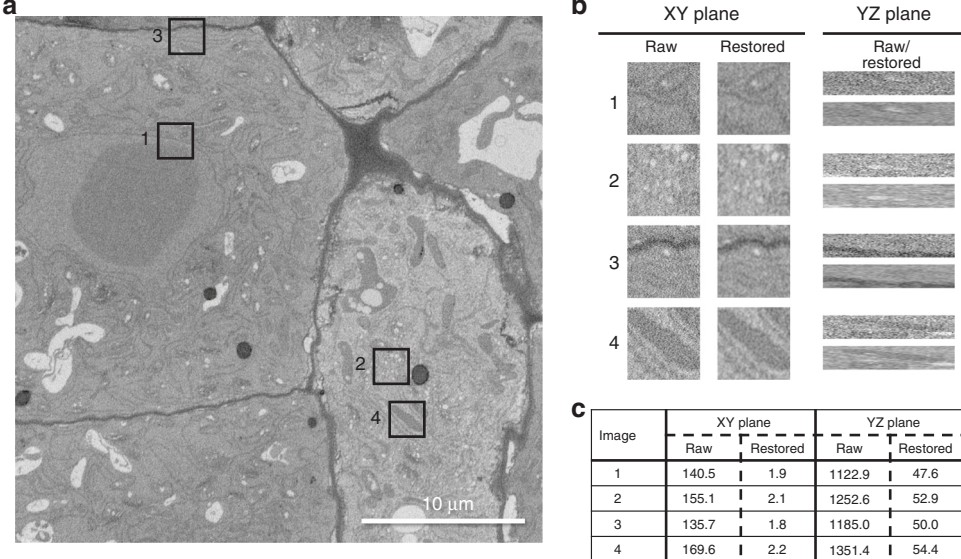

**Fig. 3 An SBF-SEM section of an Arabidopsis thaliana root tip and four ROIs that show the denoising performance of DenoisEM. a** The original image and four annotated ROIs and **b** a qualitative comparison of the original and denoised cross sections. For each ROI we show an XY and YZ section to illustrate that the image quality also improves along the Z direction, even though DenoisEM restores each XY slice independently. **c** For each ROI, we provide an estimation of the noise standard deviation $\sigma_n$ ref. [51] in the raw and denoised patch to illustrate the image quality quantitatively. Note that the noise level decreases by almost two orders of magnitude.

Figure 3c table:

| Image | XY plane | | YZ plane | |
|---|---|---|---|---|
| | Raw | Restored | Raw | Restored |
| 1 | 140.5 | 1.9 | 1122.9 | 47.6 |
| 2 | 155.1 | 2.1 | 1252.6 | 52.9 |
| 3 | 135.7 | 1.8 | 1185.0 | 50.0 |
| 4 | 169.6 | 2.2 | 1351.4 | 54.4 |

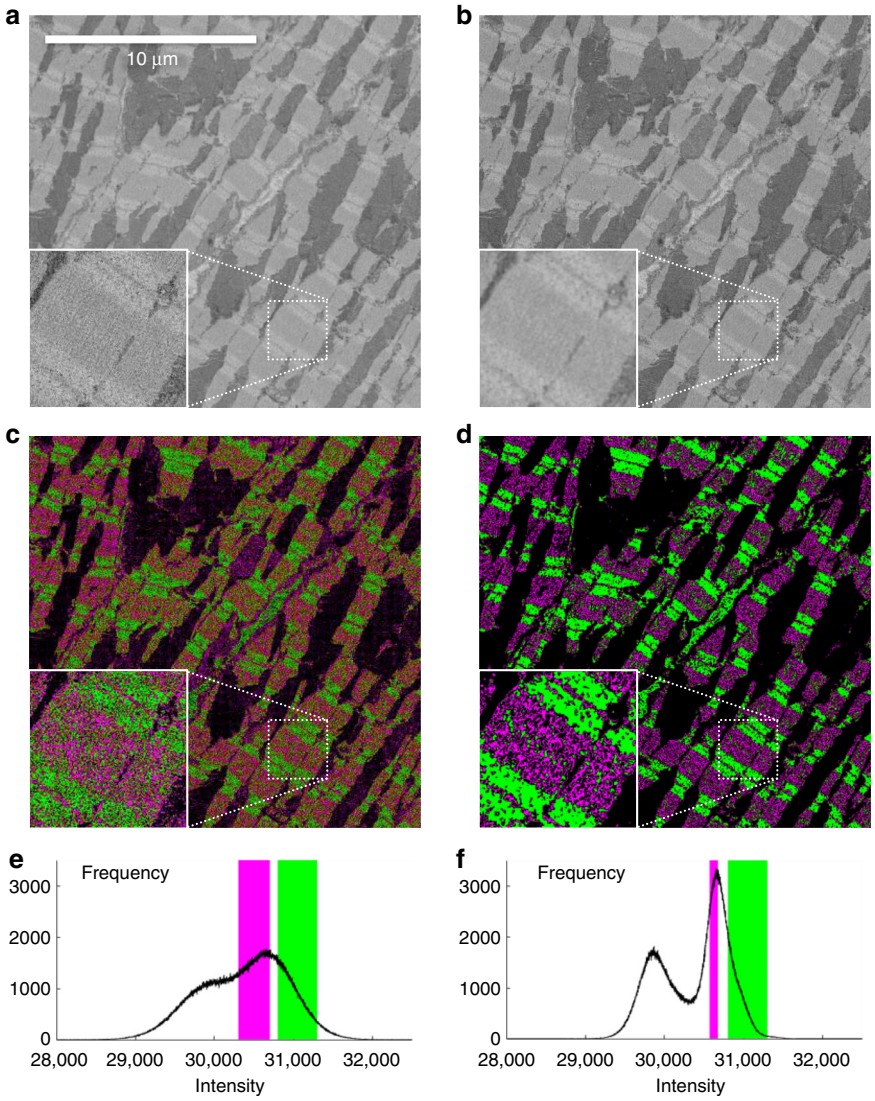

**Fig. 4 Influence of denoising on thresholding segmentation of A/I-bands in mouse heart tissue SBF-SEM data. a** One image of an SBF-SEM datastack is shown. The inset shows a zoom in the region indicated by the box. **b** The same image is shown after applying denoising. For visualization, thresholding was applied for two ranges of intensity values and a magenta mask was used to indicate A-bands, a green mask to indicate I-bands. **c, d** show the masks for the raw and denoised data, respectively. **e, f** show the intensity histograms that correspond to images (**a**) and (**b**), with indication in magenta and green of the threshold values that were used to generate images (**c**) and (**d**).

499 slicings are done. Each slicing event takes 18 s, and the acquisition of one image takes 6'40" at a dwell time of 4 μs, leading to a general acquisition and slicing time of 58 h. If DenoisEM is applied to improve image quality, and 1-μs dwell time can be used, one image can be acquired in 1'40" and the whole data set can be acquired in 16 h. Note that the additional processing in the latter case is no bottleneck due to GPU acceleration. We employed Tikhonov deconvolution, which required only slightly more than 5 min on a basic laptop workstation with an NVIDIA Quadro P2000 GPU. This means that our plugin is able to accelerate the throughput of EM imaging by a factor of 3.5 without sacrificing image quality.

Note that an alternative to increase the acquisition throughput is to employ larger pixel sizes. However, this results in a significant loss of resolution. DenoisEM is provided with state-of-the-art denoising and deconvolution algorithms that reduce noise without significantly affecting image sharpness. To show this, we have acquired the same image as in Fig. 5d with a pixel size of 20 nm (i.e., a fourfold imaging acceleration factor, similar to Fig. 5b). To analyze the resolution/sharpness of the image, we inspect the

Fourier spectrum (see Supplementary Fig. 11). The corresponding spectrum shows significantly less high-frequency components, in comparison with our proposed approach that includes DenoisEM. This becomes even more clear after removing the low-frequency components of the spectrum and applying the inverse transform to the image domain. We note that the resulting edge map contains significantly more structural detail in the proposed method, whereas the image sampled at 20-nm pixel size is less sharp and contains more noise.

**Improved segmentation quality and faster image analysis.** Mouse heart tissue was prepared as in Vanslembrouck et al.[49] and imaged using FIB-SEM. In this particular data set, the lateral view shows a transverse filament section (Fig. 6a). The noise in the image is removed with DenoisEM by applying the nonlocal means algorithm with damping parameter $h = 0.23$, half window size $B = 6$ and half search window size $W = 9$ (Fig. 6d). For both the raw and denoised data, a rendering mask was created by intensity-based thresholding (Fig. 6b, e) with the ImageJ 3D

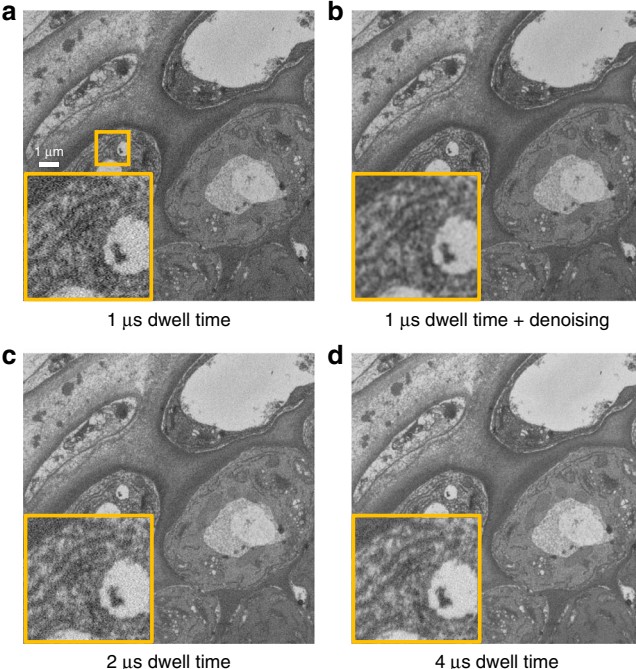

**Fig. 5 A comparison of short dwell times (+ denoising) and long dwell times.** An image acquired at a short dwell time, (**a**) before and (**b**) after denoising, and images acquired at (**c**) two and (**d**) four times longer dwell times. Notice how the noise level decreases and visualization of ultrastructure improves as the dwell time increases. By including denoising in the pipeline, we can benefit from both high-quality visualization and short dwell times.

Viewer (Fig. 6c, f). Denoising was crucial for the 3D visualization of the filaments as separate objects. For counting these, we chose the corresponding region in the raw and denoised image (Fig. 6g, i). Two experts counted the number of filaments in these ROIs three times, using the ImageJ Cell Counter plugin. Cell Counter allows a user to identify an object with a mouse-click and indicates each click on the image with a yellow pixel. When the experts performed the counting, it was more straightforward to recognize individual filaments in the denoised image, and the results in Fig. 6k showed a lower count in the raw image, as compared with the denoised image. As an alternative for counting objects, the ImageJ Analyze Particles was used on a thresholded image. With this automatic segmentation tool, the number of counted objects in the denoised image corresponds to the manual counting, especially when applying a watershed filter prior to Analyze Particles (Fig. 6j, k). The same analysis on the raw image, resulted in values that are off by a factor of more than 1.7 compared with manual counting (Fig. 6h, k). This demonstrates that the raw images were not suited for automatic object counting, and that the introduction of expert-guided denoising can prepare the images for an automatic analysis workflow. Also timewise, there was a clear benefit, since the analysis by thresholding, watershed, and the Analyze Particles tool required 20 s per image on average, while manual counting required at least 2.5 min.

To illustrate how our plugin can improve automated segmentation quality, we have performed experiments with the CREMI challenge data set (https://cremi.org/) where volumes of serial section TEM of the adult fly brain are considered for neuron reconstruction. We extracted the outer neuron membranes by computing the edges between the different neurons (using a Sobel filter) and dilating these edges by five pixels, the mean membrane width in the data set. We use the pixel classifier

from the ilastik framework[50] to predict neuronal membranes. The prediction uses intensity, edge, and texture features and a random forest pixel classifier combined with a small number of annotated pixels: ~4000 and 7000 membrane and background pixels, respectively (see top-left image in Fig. 7). To evaluate the influence of noise and denoising, we simulate noise of variable levels (i.e., $\sigma_n = 0.01, 0.05, 0.1, 0.2$) on the original (approximately noise-free) data and denoised these noisy data sets with the nonlocal means algorithm with a half window size $B = 4$, half search window size $W = 5$ and damping parameter $h = 0.05, 0.15, 0.4, 0.7$, respectively. Expert intervention was necessary to finetune the damping parameter such that neuronal membranes were preserved for each noise level. Segmentation is performed on both the noisy and denoised data sets using the same annotated pixels. Figure 7 shows the generated probability maps and segmentation masks, and Fig. 8 illustrates the quantitative results in relation to the noise level by means of the Dice coefficient ($D = 2\frac{|P \cap S|}{|P| + |S|}$, where $P$ and $S$ are the set of predicted and true foreground/background pixels, respectively). The results show that even a more advanced segmentation algorithm such as random forests pixel classification is not robust enough, as segmentation quality increases whenever denoising is performed as a pre-processing step. This is especially notable for higher noise levels, which is often the case when a short acquisition time was used (e.g., due to time or sample restrictions). Given that the pixel dwell time is inversely related to the noise level $\sigma$ in the image, it can be inferred from Fig. 8 that equal segmentation performance is achievable by accelerating the image acquisition by a factor of 20 ($\sigma = 0.01$ vs. $\sigma = 0.2$) and including denoising. The segmentation results even improved for a tenfold acquisition acceleration ($\sigma = 0.01$ vs. $\sigma = 0.1$) when denoising is included. Practically speaking, this implies that acquisition times can be reduced from hours to minutes without sacrificing segmentation quality. Note that the denoising itself does not require a significant amount of overhead: e.g., the processing time for nonlocal means on a single CREMI data set volume ($1250 \times 1250 \times 125$ pixels) requires less than a minute with a modern GPU.

**Performance at low latency and easily extensible**. In order to allow for a user in the loop, it is necessary to compute restoration results at a low latency, so that the user can test different parameter settings as fast as possible. The computational backend of DenoisEM relies on a freely available programming framework called Quasar[30], which reduces the complexity of heterogeneous programming on CPUs and GPUs to programming in a high-level language like MATLAB/Python without significant runtime influence. Quasar is therefore ideal for development of novel image processing algorithms, as prototyping is accelerated by parallel computing on the GPU and/or CPU.

As the plugin's host application (ImageJ) is implemented in the Java programming language, we developed a bridge between Java and Quasar to connect front and backend. This bridge uses the Java Native Interface (JNI) to wrap Quasar objects and functions in Java equivalents, providing an abstraction layer that separates the complexity of the Quasar C++ API from the DenoisEM plugin. The bridge is DenoisEM-agnostic and by itself a useful general building block for other applications wishing to leverage Quasar from within a Java environment. The DenoisEM plugin and the source code for the plugin, the Java-Quasar bridge and the denoising algorithms are freely available for noncommercial use.

We have compared the Quasar algorithm implementations available in DenoisEM to existing open-source implementations, which are typically not GPU accelerated. Figure 9 shows a

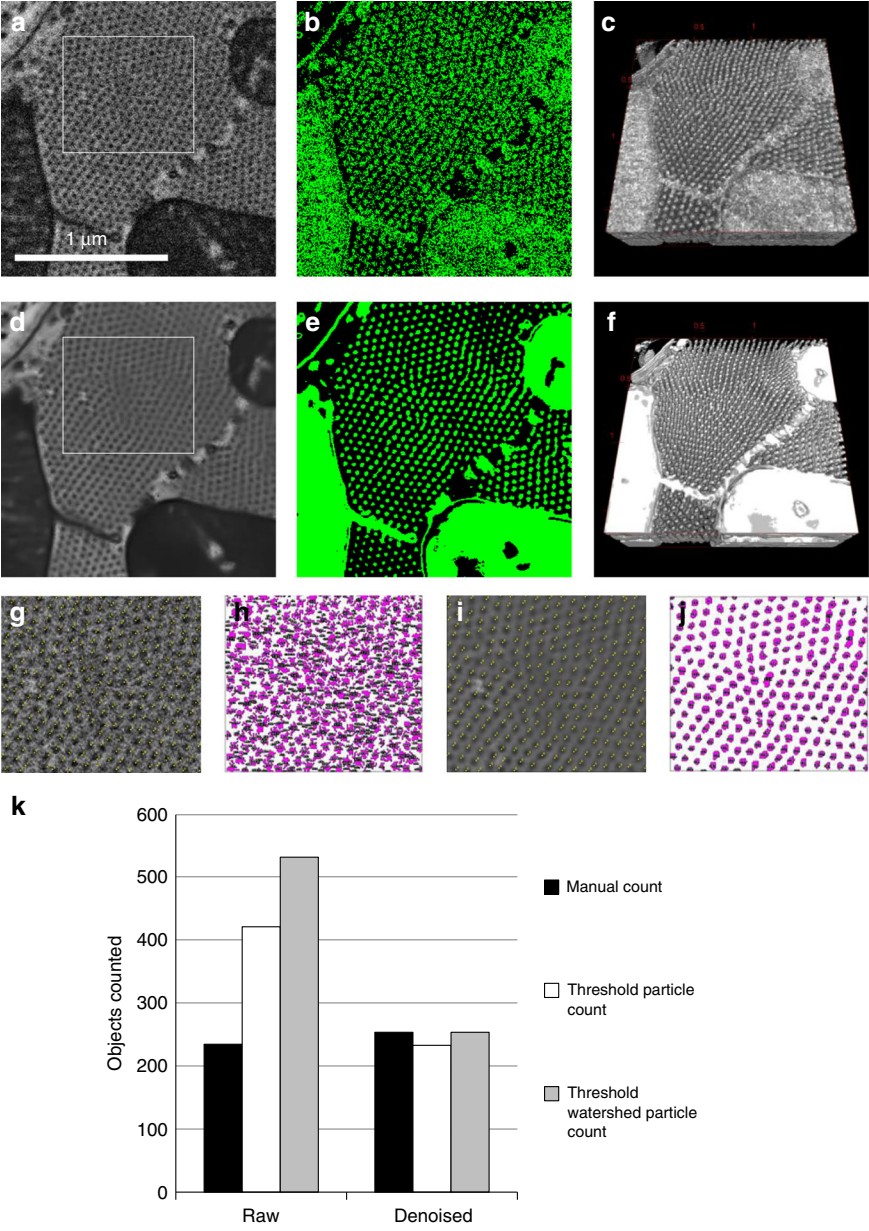

**Fig. 6 Denoising influence on sarcomere segmentation and counting in mouse heart tissue FIB-SEM data.** A 3D ROI from a FIB-SEM data set, acquired at isotropic 5-nm resolution, from mouse heart tissue was used for denoising. **a** An XY view of 330 × 300 pixels of the raw image. **b** The image was used for creating a mask (in green) by intensity thresholding. **c** This mask was used for 3D rendering with the ImageJ 3D Viewer. **d** The data were denoised with DenoisEM, using the NLM algorithm, and **e**, **f** segmented by thresholding. **g** The ROI indicated in panel (**a**) was used for manual counting of sarcomeres with the ImageJ Cell Counter Plugin. **h** Each item counted is indicated with a magenta dot. Panels (**i**) and (**j**) show the counting results on the denoised data. **k** A graph showing the number of sarcomeres counted, either manual by three different individuals or by the ImageJ Analyse Particles tool on a thresholded image, with or without watershed.

comparison for bilateral filtering[41], anisotropic diffusion[38], BLS-GSM[14], and nonlocal means denoising[16] (results of the remaining methods can be found in the Supplementary Fig. 13). The results show that the Quasar implementations are one to two orders of magnitude faster than the classical CPU-based implementations. Considering a latency threshold of 100 ms (i.e., 10 FPS), this allows real-time processing of megapixel images for the bilateral filter, anisotropic diffusion and the more recently proposed nonlocal means algorithm. State-of-the-art denoising algorithms such as BLS-GSM used to take seconds to minutes for only a few megapixel sized images, whereas our Quasar implementation requires up to 1 s for 16 megapixel images. This allows for much faster parameter tuning by the expert. Note that this acceleration

can also be obtained by incorporating e.g., CUDA libraries in the baseline implementations; however, this approach would require more development time and we believe that the high-level nature of Quasar is more scalable to this end. The high-level Quasar API has been demonstrated to support rapid software development without compromising performance in direct comparison with other popular GPU development platforms such as CUDA (see the work of Goossens et al.[30]). In addition, we can observe that the obtained GPU speedups increase for larger inputs, which is desirable for large-scale computing. This is due to the fact that more pixels can be processed in parallel, and bounded by the amount of cores in the available GPU hardware.

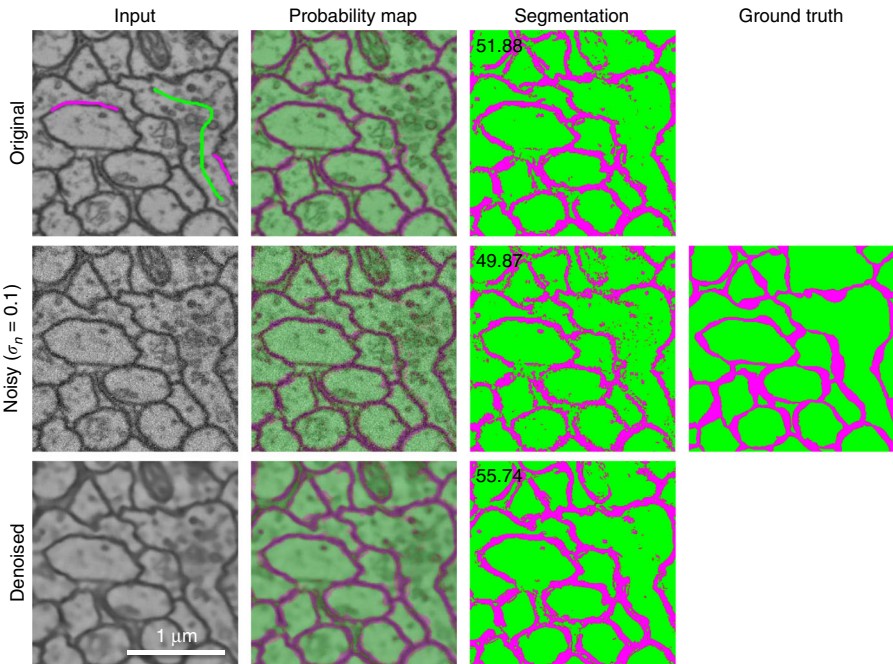

**Fig. 7 Influence of denoising on pixel-classification-based membrane segmentation in TEM data.** The first column shows the input, the top image additionally shows a fraction of the labels that were used for training. The second column shows the probability output map of the random forest pixel classifier from ilastik. The third column shows the segmentation result by thresholding the probability maps, and the fourth column the ground truth segmentation (the Dice coefficient is shown in the upper left corner). Noise artifacts are clearly visible in the segmentation and can be avoided by denoising as a pre-processing step. Notice that denoising can even improve the segmentation result.

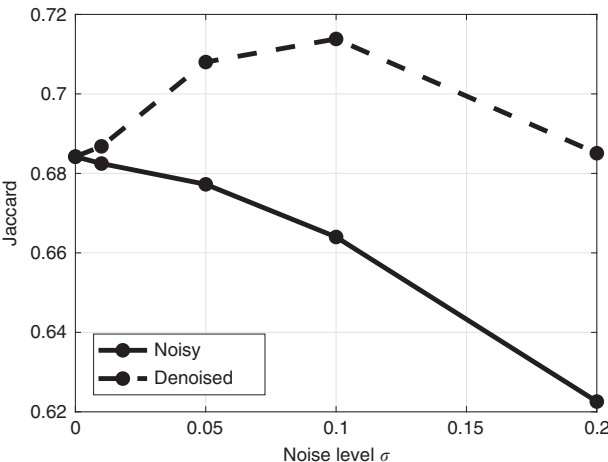

**Fig. 8 Segmentation quality of the automated membrane segmentation in the CREMI data set using ilastik.** As the noise level in the data increases, the segmentation quality is significantly influenced whereas denoising pre-processing stabilizes (and even improves) the segmentation quality.

## Discussion

There is an increasing need for higher throughput in volumetric electron microscopy (EM) imaging. Faster acquisition typically results in a higher noise level, which can impede ultrastructure analysis. Consequently, to scale up 3D EM image acquisition, there is a growing demand for fast and effective denoising algorithms. Moreover, as the final purpose of 3D EM imaging can be diverse (e.g., visualization, segmentation, counting, etc.), there is no one-fits-all algorithm. This can be seen by the plethora of algorithms that are currently applied in many 3D EM applications[20–25].

In this work, we present DenoisEM, an interactive and GPU accelerated denoising tool for volumetric data (particularly 3D EM), accessible through ImageJ. The broad range of denoising possibilities offered in one solution, make DenoisEM very versatile. Parameter tuning is efficient due to the GPU accelerated backend, which allows for visualization of different parameter settings at low latency. Reproducibility is also guaranteed by saving the algorithm parameter settings as metadata. This can be useful in a later stage, e.g., for other data sets that require exactly the same processing.

We show that we can increase the throughput of 3D EM by a factor of 4 without affecting image quality.

We validated the potential improvements that DenoisEM can provide in 3D EM image interpretation by denoising SBF-SEM image data of an Arabidopsis thaliana root tip. Experts confirmed that structures such as the nuclear membrane and endoplasmic reticulum were easier to recognize in the denoised data by optimally tuning the denoising parameters. As a second use case, we used DenoisEM on noisy SBF-SEM data of mouse heart tissue and improved the visualization of sarcomeres as the A and I-bands were better separated.

We assessed the effect of denoising on 3D EM data as a processing step prior to automated segmentation, both intensity-based thresholding and pixel-level higher-order feature classifiers. An interesting conclusion is that segmentation quality does not significantly decrease for denoised noisy inputs compared with those that are noise-free. Consequently, the acquisition time can be shortened to increase throughput or avoid overexposure, without significantly affecting subsequent segmentation results obtained by classical methods, such as thresholding or pixel classification.

An important note is that most denoising algorithms are limited in performance by a trade-off between noise reduction and edge blurring. Blur affects the resolving power of ultrastructure boundaries and is therefore not desired in most

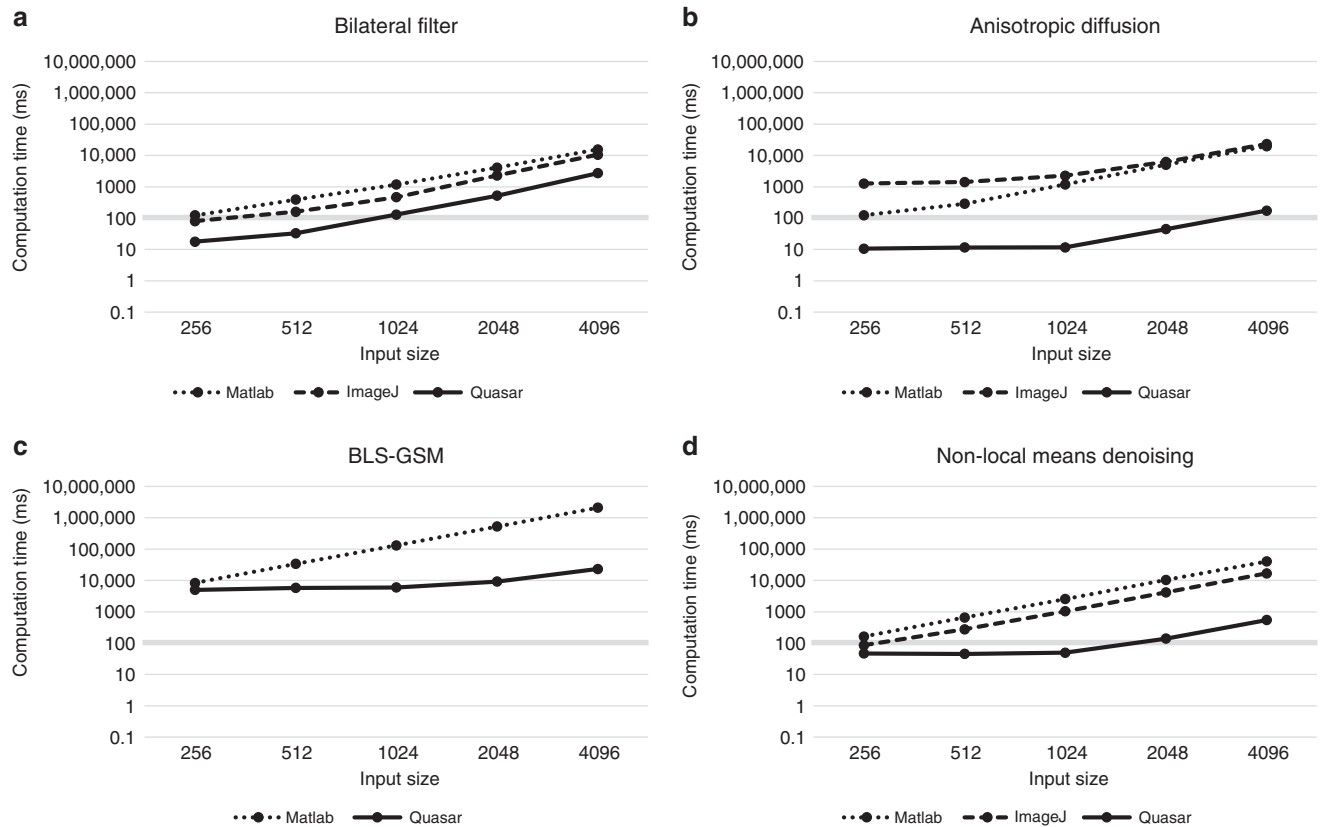

**Fig. 9 Computational performance comparison of Quasar-based and alternative available implementations.** Absolute computational performance (in milliseconds) of (**a**) bilateral filtering[41], (**b**) anisotropic diffusion[38], (**c**) BLS-GSM[14], and (**d**) nonlocal means denoising[16] for different input sizes. A comparison is made between our proposed GPU-based Quasar framework and alternative implementations that are available to the scientific community: bilateral filter (ImageJ[52] and MATLAB[53] based), anisotropic diffusion (ImageJ[54] and MATLAB[55] based), BLS-GSM (MATLAB[56]) and nonlocal means denoising (ImageJ[57] and MATLAB[58] based). For each algorithm, we consider inputs of $256^2$, $512^2$, $1024^2$, $2048^2$, and $4096^2$ pixels. In general, our Quasar implementation performs 10 to 100 times faster compared with the existing software packages. Notice that the obtained speedup increases as the input image size increases due to the fact that GPUs are able to process more pixels in parallel.

experiments. The trade-off is typically moderated through (a combination of) parameters and should therefore be tuned properly in order not to "overprocess" data sets. We believe that validation of the optimal parameter settings can only be performed by experts. DenoisEM allows for fast comparison of different parameter settings and state-of-the-art algorithms through its accelerated backend.

We believe that DenoisEM is a plugin, accessible to the life science community, that can significantly improve the quality and throughput of large-scale 3D EM imaging. Although we focus our plugin on 3D EM data, it can also be used for any (2D/3D) grayscale image producing modality (e.g., medical or astronomical data). Future work will focus on predictive parameter optimization based on regression models and correlation analysis between the parameters of different algorithms. By keeping track of the EM metadata (e.g., modality, cell type, acquisition time, etc.) and the eventual objective (segmentation or visualization), we believe that parameter optimization can be further automated. In addition, we will also extend the DenoisEM framework to multichannel image data to include the light microscopy community. Moreover, the high-level nature of Quasar allows us to easily implement novel restoration algorithms (e.g., based on convolutional neural networks) that could further improve the restoration quality.

## Methods

**Sample preparation**. Animals were sacrificed for dissection. C57BL/6 wild-type female mice of 8 weeks old were maintained in standard specific pathogen-free (SPF) housing according to the European rules on animal welfare at the animal

facility of the Center for Inflammation Research at VIB (Ghent, Belgium). Mice were anesthetized by intraperitoneal injection of ketamine/xylazine (70 mg of ketamine and 10 mg of xylazine per kg of body weight) and perfused, first with PBS containing heparin (20 units/ml) for 2 min, followed by 2% paraformaldehyde (PFA; AppliChem, Darmstadt, Germany), 2.5% glutaraldehyde (Electron Microscopy Sciences (EMS), Hatfield, PA USA) in 0.15 M cacodylate buffer (Sigma-Aldrich, Overijse, Belgium), pH 7.4, for 10 min. Next, heart muscle tissue was isolated and fixed overnight using 2% PFA, 2.5% glutaraldehyde in 0.15 M cacodylate buffer, pH 7.4. Samples were thoroughly washed in 0.15 M cacodylate buffer, pH 7.4, before small blocks were dissected to proceed with the staining protocol. Post-fixation was performed by incubating tissue blocks in 1% osmium (EMS), 1.5% potassium ferrocyanide (EMS) in 0.15 M cacodylate buffer, pH 7.4.

For SBF-SEM, post-fixation with osmium was followed by incubation in 1% thiocarbohydrazide (TCH; EMS) and subsequent washes in double-deionized water (ddH2O). Next, a second incubation in 2% osmium in ddH₂O was performed. Both TCH and the second osmication were repeated after this. The samples were then washed in ddH₂O and placed in 2% uranic acetate (UA; EMS). After the following washing step, Walton's lead aspartate staining was performed for 30 min at 60 °C. For this, a 30 mM l-aspartic acid solution was used to freshly dissolve lead nitrate (20 mM, pH 5.5) just before incubation.

For FIB-SEM, the fixed tissue blocks were washed in ddH₂O for four consecutive steps, refreshing the ddH₂O after every step. Next, incubation in 1% osmium in ddH₂O was followed by washing in ddH₂O, incubation in 1% UA and again washing steps with ddH₂O.

After the final washing steps, samples for both FIB-SEM and SBF-SEM were dehydrated using solutions of 50, 70, 90, and twice 100% ethanol. Samples were then placed in 100% acetone and embedded in Spurr's resin (EMS) by incubation in 50% Spurr's in acetone, followed by four incubations in 100% Spurr's. Polymerization was done overnight at 60 °C. Except for the Walton's lead staining, all steps were performed using a Pelco Biowave Pro Microwave Tissue Processor (Tedpella Inc, Redding, CA, USA).

For SBF-SEM, the sample was mounted onto an aluminum pin, trimmed into a pyramid shape using an ultramicrotome (Leica, Ultracut) and the block surface was trimmed until smooth and at least a small part of tissue was present at the block

face. Next, samples were coated with 5 nm of platinum (Pt) in a Quorum Q 150T ES sputter coater (www.quorumtech.com). The aluminum pins were placed in the Gatan 3View2 in a Zeiss Merlin SEM. For FIB-SEM samples were mounted on Aluminum stubs (EMS) and coated with 10 nm of Pt.

**Image acquisition**. For SBF-SEM acquisitions, a Zeiss Merlin with Gatan 3View2 was used. Acquisition parameters for images of heart tissue were acceleration voltage of 1.7 kV and 100 pA, dwell time of 1 μs. Images were collected at 8000 × 8000 pixels with a pixel size of 12.44 nm, and slicing was done at z-steps of 50 nm. For image reconstructions, a ROI of 1500 × 1500 pixels was chosen and its 101 consecutive images. Acquisition parameters for images of Arabidopsis thaliana root tips in Fig. 3 were collected at acceleration voltage of 1.6 kV and 80 pA, dwell time of 1 μs. The pixel size was 13 nm, and the slice thickness 70 nm. For Fig. 5, a root tip was imaged at 1.6 kV and 100 pA. A region of 2000 × 2000 pixels with 8-nm pixel size was acquired with increasing dwell times of 1, 2, and 4 μs. FIB-SEM imaging of murine heart tissue samples was done with a Zeiss Crossbeam 540 at 5-nm pixels and slicing at 5-nm sections.

**Implemented restoration methods**. In this section, we will give a brief overview of the implemented algorithms. This includes the median absolute deviation (MAD) noise estimator, least-squares parameter estimator, Gaussian filtering (GF), wavelet thresholding (WT), anisotropic diffusion (AD), bilateral filtering (BF), Tikhonov denoising/deconvolution (TIK-DEN/DEC), total variation denoising (TV-DEN), Bayesian least-squares Gaussian scale mixtures (BLS-GSM), and nonlocal means denoising/deconvolution (NLM/NLMD). The restoration methods are complementary in terms of computation time and restoration performance: e.g., algorithms such as BLS-GSM and NLMD are computationally more intensive than, for example, GF and AD, but the former are more likely to outperform the latter. This trade-off between computation time and image quality can be tuned by the expert. In Supplementary Figs. 2–10, we illustrate the influence of the implemented methods and their respective parameters on the restored image.

We will denote 3D images as vectors by stacking the pixels using slice-by-slice raster scanning ordering. More specifically, the acquired image $\mathbf{y} \in \mathbb{R}^N$ (where $N$ is the number of voxels in the image) is assumed to be degraded by blur and additive noise:

$$\mathbf{y} = \mathbf{Hx} + \mathbf{n}, \tag{1}$$

where $\mathbf{x} \in \mathbb{R}^N$ is the underlying, degradation-free image, $\mathbf{H} \in \mathbb{R}^{N \times N}$ represents the microscope point-spread function (PSF), and $\mathbf{n} \in \mathbb{R}^N$ is a stochastic, noise term. The noise term is assumed to be mean-zero and with a constant variance, i.e., $[\mathbf{C}]_{i,i} = \sigma^2$ (where $\mathbf{C}$ is the noise-covariance matrix).

Note that all the described denoising algorithms are independent of the dimensionality of the data set. Consequently, all of these methods can be implemented in 3D, which can improve restoration quality. However, this comes at the cost of memory and compute time. As an illustration, we have implemented the Gaussian filter, bilateral filter, and tikhonov denoising in 3D and concluded that the small increase in denoising performance was not convincingly worth the computational cost (typically 50% more) and memory overhead (typically overflow at 0.5 GV images with a 4 GB GPU). For more details on this comparison, we refer to Supplementary Fig. 14. For this reason, the described methods are implemented in 2D, and volumes are processed in a slice-by-slice fashion. We do, however, consider efficient 3D implementations as future work.

Noise estimation: This involves the task of estimating the noise variance $\sigma^2$ based on the acquired image $\mathbf{y}$. The method that is implemented in our plugin is the median absolute deviation (MAD) noise estimator:

$$\hat{\sigma} = \text{med}(|\mathbf{y} - \text{med}(\mathbf{y})|), \tag{2}$$

where med($\cdot$) denotes the median operator. The absolute difference of $\mathbf{y}$ and its median value provides a pixel-wise, noise-robust measure of how much the signal deviates from its expected value. Taking the median over these values therefore gives a robust estimation of the noise standard deviation over the complete image $\mathbf{y}$.

Blur estimation: This involves the task of quantifying the amount of blur in an image, i.e., a sharp image would have a metric of 0, and this value increases to 1 as the amount of blur increases. The implemented approach explains blur as the difference between the pixel variation of the original image $\mathbf{y}$ and a smoothened version $\mathbf{y}' = \mathbf{S}_x \mathbf{y}$, where $\mathbf{S}_x$ is a horizontal smoothing kernel. The pixel variation is defined as:

$$\mathbf{v}_x = \max(0, \mathbf{D}_x \mathbf{y} - \mathbf{D}_x \mathbf{y}'), \tag{3}$$

where $\mathbf{D}_x$ expresses the variation along the x-axis (e.g., a first-order derivative approximation). These pixel-wise variations are aggregated to a single value and normalized as follows:

$$b_x = \frac{\sum_i [\mathbf{D}_x \mathbf{y}]_i - \sum_i [\mathbf{v}_x]_i}{\sum_i [\mathbf{D}_x \mathbf{y}]_i}. \tag{4}$$

This value expresses the blur level along the horizontal direction. However, a similar approach can be used to extract a blur estimation $b_y$ along the vertical direction. The final blur metric is simply the maximum of both values.

Parameter estimation: Consider an algorithm $\mathbf{f}_\theta(\cdot)$ with parameters $\boldsymbol{\theta} \in \mathbb{R}^p$ that maps a noisy image $\mathbf{y}$ onto a noise-free estimation $\hat{\mathbf{x}} = \mathbf{f}_\theta(\mathbf{y})$. Based on the estimated noise level $\hat{\sigma}$ (see Eq. (2)), we estimate the optimal parameter settings through parameterized polynomial expressions w.r.t. a training data set of noise-free benchmark EM images $\mathbf{x}_k$, for $k = 1, \ldots, K$ ($K = 100$ in our case). These images were degraded with Gaussian noise of different levels $\sigma_m$ resulting in noisy images $\mathbf{y}_{k,m}$. The optimal parameters $\boldsymbol{\theta}_m \in \mathbb{R}^p$ were determined for each noise level $\sigma_m$ in least-squares sense:

$$\boldsymbol{\theta}_m = \arg\min_{\boldsymbol{\theta}} \sum_k \left\| \mathbf{f}_\theta(\mathbf{y}_{k,m}) - \mathbf{x}_k \right\|^2 \tag{5}$$

Next, the polynomial coefficient parameters $\mathbf{a}_j \in \mathbb{R}^q$ were optimized with a least-squares fit:

$$\mathbf{a}_j = \arg\min_{\mathbf{a}} \sum_m \left( [\boldsymbol{\theta}_m]_j - \sum_i [\mathbf{a}]_i \sigma_m^i \right)^2 \tag{6}$$

Finally, the estimated parameters that correspond to the estimated noise level $\hat{\sigma}$ are computed as:

$$\left[ \hat{\boldsymbol{\theta}} \right]_j = \sum_i [\mathbf{a}_j]_i \hat{\sigma}^i. \tag{7}$$

In practice, we concluded that, depending on the algorithm, a linear or quadratic polynomial fit ($q = 1, 2$) approximates the optimal parameters (in least-squares sense) sufficiently close (see Supplementary Figs. 15, 16).

Gaussian filter: This is a special case of linear, smoothing filters which combine pixel values linearly in a local neighborhood in order to restore a pixel value. Practically, this comes down to a convolution of the noisy image $\mathbf{y}$ with a convolution kernel $\mathbf{G}$ (which is Gaussian in this case).

Wavelet thresholding: Wavelet transforms separate image information across multiple frequency scales and magnitudes. Noise tends to be spread among all the transformed coefficients whereas the coefficients that represent actual discontinuities typically stand out. Therefore, a popular method to reduce the amount of noise in an image is to reduce the magnitude of the transformed coefficients, this is also known as wavelet shrinkage. More specifically, the restored image is found by transforming the image $\mathbf{y}$ to the wavelet domain, shrink the noisy coefficients and transform back to the spatial domain:

$$\hat{\mathbf{x}} = \mathbf{W}^H(\tau(\mathbf{Wy})), \tag{8}$$

where $\mathbf{W}$ represents the used wavelet transform and $\mathbf{W}^H$ is its Hermitian transpose. The shrinkage function $\tau$ is typically the soft-thresholding operator:

$$\tau(\mathbf{z}) = \text{sign}(\mathbf{z}) \max(|\mathbf{z}| - T, 0), \tag{9}$$

where all the functions operate component-wise and $T$ is a thresholding parameter.

Anisotropic diffusion: The anisotropic diffusion filter, commonly used in the context of EM image restoration, introduces nonlinearity by describing linear filters in a partial differential equation (PDE) domain and extending it to a nonlinear case. The true image $\mathbf{x}$ is embedded in a family of images $\mathbf{x}_t$, obtained by convolving the image $\mathbf{x}$ with Gaussian filters with a variance that increases with $t$. This diffusion process can be described by the so-called linear diffusion heat equation:

$$\frac{\partial \mathbf{x}_t}{\partial t} = \nabla \cdot (\nabla \mathbf{x}_t), \tag{10}$$

where $\nabla$ represents the gradient operator with respect to the spatial coordinates. This isotropic diffusion method ignores edge information and consequently blurs edges. The anisotropic diffusion filter mitigates this by integrating a gradient magnitude weighting function:

$$\frac{\partial \mathbf{x}_t}{\partial t} = \nabla \cdot \left( c(\|\nabla \mathbf{x}_t\|_2) \nabla \mathbf{x}_t \right). \tag{11}$$

Two nonlinear gradient regularization functions are commonly used:

$$c(s) = \exp\left( -\left(\frac{s}{\kappa}\right)^2 \right), \tag{12}$$

$$c(s) = \frac{1}{1 + \left(\frac{s}{\kappa}\right)^2}, \tag{13}$$

where $\kappa > 0$ is a parameter that offers a trade-off between noise reduction and edge preservation. The idea is that the function $c(\cdot)$ returns small values for large gradient magnitudes and vice versa such that edges are less diffused (i.e., blurred) and only in the direction of the edge (e.g., to avoid that horizontal edges will be blurred in the horizontal direction).

Bilateral filter: It is argued that local linear filters (such as the Gaussian filter) tend to oversmooth image edges and require nonlinearities in order to obtain a better restoration estimate. In former EM research, the bilateral filter is used as an alternative: this is a locally adaptive spatial filter that avoids oversmoothing by averaging less aggressively along edges. More specifically, the restored $i$th pixel $[\hat{\mathbf{x}}]_i$

is computed as:

$$[\hat{\mathbf{x}}]_i = \sum_j f_{\text{int}}(|[\mathbf{y}]_i - [\mathbf{y}]_j|) f_{\text{sp}}(\|\mathbf{p}_i - \mathbf{p}_j\|_2)[\mathbf{y}]_j, \quad (14)$$

where $f_{\text{int}}, f_{\text{sp}} : \mathbb{R} \to \mathbb{R}$ are kernel functions that weigh the intensity and spatial distance, respectively, and $\mathbf{p}_i \in \mathbb{R}^3$ represents the 3D spatial position vector that corresponds to the index $i$. Similar to the Gaussian filter, pixels $[\mathbf{y}]_j$ nearby the reference pixel $[\mathbf{y}]_i$ will be assigned larger averaging weights through $f_{\text{sp}}$. However, pixel intensities that differ very much from the reference pixel (typically edges) are assigned low-weight values through $f_{\text{int}}$, which leads to less blurring along edges.

Tikhonov deconvolution: Tikhonov restoration exploits the fact that images generally consist of many smooth regions separated by a much smaller amount of edges. As a consequence, the total edge magnitude in the restored image should be penalized, in other words:

$$\hat{\mathbf{x}} = \arg\min_{\mathbf{x}} \|\mathbf{y} - \mathbf{H}\mathbf{x}\|_2^2 + \lambda\|\mathbf{L}\mathbf{x}\|_2^2, \quad (15)$$

where $\mathbf{L}$ is typically a matrix that acts as a gradient or Laplacian operator in order to quantify edges in the image (we use the Laplacian).

Total variation deconvolution: The total variation prior (TV) assumes that natural images $\mathbf{x}$ should consist of flat regions delineated by a relatively small amount of edges. Mathematically, this is expressed as minimizing the total variation of the image:

$$\hat{\mathbf{x}} = \arg\min_{\mathbf{x}} \|\mathbf{y} - \mathbf{H}\mathbf{x}\|_2^2 + \lambda \sum_i \sqrt{[\mathbf{D}_x\mathbf{x}]_i^2 + [\mathbf{D}_y\mathbf{x}]_i^2 + [\mathbf{D}_z\mathbf{x}]_i^2}, \quad (16)$$

where $\mathbf{D}_x, \mathbf{D}_y$, and $\mathbf{D}_z$ are matrices that express the variation of the image along the $x$, $y$, and $z$-axis, respectively (e.g., first-order derivative approximations).

BLS-GSM: The BLS-GSM method decomposes the image into $J$ scales and $K$ oriented pyramid subbands, denoises the high-pass subbands and inverts the pyramid transform. An $M \times M$ neighborhood around a reference coefficient $[\mathbf{v}]_c$ of a subband is considered and represented as a vector $\mathbf{v}$, by column stacking. These coefficients are then modeled as Gaussian Scale Mixtures (GSM), i.e., the product of a Gaussian distributed vector $\mathbf{u} \stackrel{d}{=} \mathcal{N}(\mathbf{0}, \mathbf{C}_u)$ (where $\stackrel{d}{=}$ indicates distribution equality) and the square root of an independent, positive scalar random variable $z$:

$$\mathbf{v} \stackrel{d}{=} \sqrt{z}\mathbf{u}, \quad (17)$$

such that a noisy neighborhood patch $\mathbf{w}$ is described by:

$$\mathbf{w} = \mathbf{v} + \mathbf{n}, \quad (18)$$

where $\mathbf{n} \stackrel{d}{=} \mathcal{N}(\mathbf{0}, \mathbf{C}_n)$ are the noise coefficients. Based on this model, the reference coefficient is approximated using the Bayesian least-squares (BLS) estimator which reduces to:

$$[\hat{\mathbf{v}}]_c = \int_0^\infty \mathrm{P}(z|\mathbf{w}) \mathrm{E}\big[[\mathbf{v}]_c|\mathbf{w}, z\big] \, dz, \quad (19)$$

where $\mathrm{P}(z|\mathbf{w})$ is found with Bayes' rule, and $\mathrm{E}[\mathbf{v}|\mathbf{w}, z]$ is computed as a local linear Wiener estimate:

$$\mathrm{E}[\mathbf{v}|\mathbf{w}, z] = z\mathbf{C}_u(z\mathbf{C}_u + \mathbf{C}_n)^{-1}\mathbf{w}. \quad (20)$$

Mainly due to accurate statistical noise modeling, BLS-GSM has become the state-of-the-art in multiresolution-based denoising.

Nonlocal means denoising: Since the introduction of the nonlocal means (NLM) filter, self-similarity denoising approaches gained a lot of interest because of their high performance. More specifically, the NLM algorithm estimates the noise-free image pixels as weighted averages of acquired pixel values:

$$[\hat{\mathbf{x}}]_i = \frac{\sum_j w_{i,j}[\mathbf{y}]_j}{\sum_j w_{i,j}}. \quad (21)$$

The self-similarity constraint turns up in the way the weights $w_{i,j}$ are computed: these should be large (respectively, small) for similar (respectively, dissimilar) pixels $i$ and $j$. Pixel similarity is defined along a local pixel neighborhood, and searched for in both a local and nonlocal region:

$$w_{i,j} = \exp\left(-\frac{1}{2}\frac{\|\mathbf{y}_{N_i} - \mathbf{y}_{N_j}\|_2^2}{h}\right) \quad (22)$$

where $\mathbf{y}_{N_i}$ denotes a local neighborhood $N_i$ of the acquired pixel $[y]_i$, and $h$ is a similarity damping parameter.

It has been shown that the NLM algorithm can be equivalently expressed by means of a Bayesian estimator with nonlocal image prior. An NLM deconvolution algorithm can be derived by extending the Bayesian estimator with nonlocal prior to a deconvolution estimator:

$$\hat{\mathbf{x}} = \arg\min_{\mathbf{x}} \|\mathbf{y} - \mathbf{H}\mathbf{x}\|_2^2 + \lambda \sum_{i,j=0}^{N-1} w_{i,j}\|[\mathbf{x}]_i - [\mathbf{x}]_j\|_2^2. \quad (23)$$

where $\lambda$ is a regularization parameter and $\mathbf{H}$ is the estimated PSF of the microscope.

**Image restoration settings for the experiments**. Most image restoration methods have parameters that may affect computational performance: e.g., search windows, block sizes, number of iterations, etc. For computational comparison, we selected parameter settings that corresponded to those that were most frequently preferred by biological experts during our experiments. In particular, we only report the parameters that affect computation time:

- Gaussian filter: window size $7 \times 7$ (fixed in the plugin)
- Wavelet thresholding: 3D dual-tree complex wavelet transform, 6 scales
- Anisotropic diffusion: $N = 5$ iterations
- Bilateral filter: window size $15 \times 15$ (fixed in the plugin)
- Tikhonov denoising: $N = 10$ iterations
- Total variation denoising: $N = 100$ iterations
- BLS-GSM: $J = 3$ scales, window size $3 \times 3$ (fixed in the plugin)
- Nonlocal means denoising: half window size $B = 4$, half search window size $W = 5$
- Nonlocal means deconvolution: half window size $B = 4$, half search window size $W = 5$, $N = 20$ iterations

The algorithms were implemented in Quasar and compared to open-source MATLAB-based and ImageJ-based implementations if available. The experiments were performed with an Intel(R) Core(TM) i7-4930K CPU @ 3.40 GHz and an NVIDIA GeForce GTX 1070 GPU and repeated for various sizes of 2D input images ($256^2$, $512^2$ up to $4096^2$). For more details, we refer to Supplementary Fig. 13.

SBF-SEM images, acquired with the Zeiss Merlin and Gatan 3View2 detector, are originally generated as 16-bit images. All image restoration techniques (denoising, registration, segmentation) were performed with floating point precision. The mouse lung artery data were restored using wavelet thresholding with a threshold value of $T = 0.08$. The Arabidopsis thaliana root tip was restored using Tikhonov deconvolution with the following parameters: $\lambda = 1.5$, $\sigma = 0.31$, and $N = 86$ iterations. The murine heart tissue data set was processed with anisotropic diffusion with a step size $v = 0.07$, $N = 6$ iterations and a diffusion factor $\kappa = 0.18$. ImageJ was used to apply thresholding on the raw and denoised image. The values of the selected intensities for the A-bands (shown in red in Fig. 4) were [30300, 30700] and [30589, 30674] for the raw and denoised image, respectively. The values of the selected intensities for the I-bands (shown in green in Fig. 4) were [30800, 31300] for both the raw and denoised image. The mouse heart tissue was denoised using nonlocal means with damping parameter $h = 0.23$, half window size $B = 6$ and half search window size $W = 9$. A reference area of $340 \times 340$ pixels and 117 sections was cropped and used in local template matching for registration. Reconstruction was performed similarly as with the murine heart tissue, using threshold values from the interval [32, 78]. As a final step, a conversion to 8-bit was always done before exporting to PNG, to allow for final visualization. Graphics design was performed using Matlab, GIMP, Inkscape, and MS Office.

**Statistics and reproducibility**. All denoising and segmentation experiments were repeated for at least ten times on different regions of interest on the data, and similar results were validated. Experiments that involved counting were repeated for three times by two independent investigators, and similar results were obtained. Timing experiments were repeated for 20 times with a similar outcome as a result. Supplementary Note 2 provides a data reproducibility statement on this paper.

**Reporting summary**. Further information on research design is available in the Nature Research Reporting Summary linked to this article.

## Data availability

The DenoisEM plugin is available on our webpage (https://bioimagingcore.be/DenoisEM) and figshare (https://doi.org/10.6084/m9.figshare.9929201). A user manual is provided on the webpage (http://bioimagingcore.be/DenoisEM/user-manual.pdf, https://doi.org/10.6084/m9.figshare.9929888). The raw data that was used for this paper are located on our webpage (https://bioimagingcore.be/DenoisEM/data, https://doi.org/10.6084/m9.figshare.9929183). The source data underlying Figs. 1–9 and Supplementary Figs. 1–17 are provided as a Source Data file (https://doi.org/10.6084/m9.figshare.9929216).

## Code availability

We stimulate the community to build on our work by open-sourcing DenoisEM (https://github.com/vibbits/EMDenoising) and the Java-Quasar bridge that (https://github.com/vibbits/JavaQuasarBridge). Supplementary code for the automated parameter estimation is provided on figshare (https://doi.org/10.6084/m9.figshare.9929228).

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

## Acknowledgements

This research has been made possible by the Agency for Flanders Innovation & Entrepreneurship (Grant Number: IWT.141703), BOF (Grant number: BOF15/PDO/003) and the Flemish Government under the "Onderzoeksprogramma Artificiële Intelligentie (AI) Vlaanderen" program. We gratefully acknowledge the support of NVIDIA Corporation with the donation of the Titan X Pascal GPU used for this research. The Zeiss Merlin with 3View and Zeiss CrossBeam 540 were purchased by funding from the CLEM-grant and VIB Technology Fund.

## Author contributions

J.R., F.V., A.K., A.G. and S.L. conceived and designed the experiments. J.R., J.A., H.L. and B.G. implemented the denoising and registration algorithms. F.V. and B.G. developed the software plugin. J.R., F.V., A.K., A. G., J.A., H.L., B.G., W.P., S.L. and Y.S. analyzed the results and wrote the paper.

## Competing interests

The authors declare no competing interests.
