## [Peer Review File · Nature Communications]

Reviewers' comments:

Reviewer #1 (Remarks to the Author):

The paper describes a workflow (DenoisEM) to improve the signal to noise ratio of data sets acquired by serial block face scanning electron microscopy (SBF-SEM). The core of the workflow is an interactive graphical display that connects 'expert human eye' to the effectiveness of a set of effective noise-reducing algorithms (wavelet thresholding, anisotropic diffusion, bilateral filtering, Tikhonov denoising, total variation denoising, BLS-GSM, non-local means denoising, non-local means deconvolution). The effectiveness of the algorithms are evaluated on several large SBF-SEM data sets.

The DenoisEM workflow is a hybrid approach, bridging the application of noise-reducing algorithms to real-life datasets that required a human expert-eye to judge whether noise-reduction is indeed carried out to a level that the biological question requires.

A strong point of the manuscript is the notion that having in place a high-quality well-defined step of enhancing the signal to noise will have the effect that the data collection speed can be drastically reduced.

The implementation of DenoisEM is transparent and pragmatic, is a plugin for ImageJ and is tailored to include high performance computing using GPUs.

The manuscript is well written and the results are presented clearly. Nevertheless, the English and story flow can be improved at several instances.

I have a few general comments that would possibly improve the impact the manuscript could have on the EM field.

1. A strong point of the manuscript is the notion that by performing high quality noise filtering after data collection the actual data collection time could be improved (reduced) significantly, possibly an order of magnitude. This is important, as data collection with SBF-SEM can take hours and in most cases a day or longer.

However, this argument is not mentioned in the abstract, nor highlighted in the introduction. It is mentioned in the last paragraph of the discussion.

In order to make this point convincingly experimental data is lacking. It would be convincing if the supplemental data could be extended by a data set collected with a long dwell time (high signal to noise) and of the same specimen block a data set collected with a short dwell time (low signal to noise), e.g. with a 5 or 10 times lower dwell time. If indeed the quality of the data sets (based upon visual inspection) after processing are very similar, the argument that the throughput of data collection time is increased can be made.

It would be good if such data could be added to the supplement.

2. I miss a discussion in the manuscript on the interplay between noise filtering and resolution loss.

In general, there is an inherent unwanted effect that by applying a filter not only (statistical) noise is reduced, but also a low-pass filtering effect is introduced that could reduce the resolution of the data set (blurring the edges). An effect that is clearly illustrated in supplementary figure 8, comparing inset (h) with inset (b).

To help the user decide in choosing the filter parameters, I believe an addition to the graphical

user interface is necessary that indicates the effect the filter will have on the resolution/sharpness of the data sets. If such an addition is not available, it will be likely that datasets are 'over processed', reducing more resolution/sharpness than necessary.

Perhaps that a pragmatic solution to providing such information would be to compute for a number of 2D SBF-SEM digital slices the power spectrum (e.g. for 10-50 slices), and average those power spectra. This would provide a good measure of the actual signal spectrum in the 2D slices of the data set. Next, one could do the same for the data sets that are processed by the noise removal filter that is used to process that data. As a measure one could provide a 'number' that indicates what percentage of the signal spectrum is retained after applying a noise removal filter. Ideally, the percentage would be '100%', but one can imagine that in practice pragmatic - but reproducible - choices can be made.

The section in the manuscript on the discussion of possible loss of resolution/sharpness by the application of the noise reduction filters should be aimed at not 'overprocessing' data.

This section should also provide a means for the user on how to make sure that not too much signal is sacrificed relative to the amount of noise that is. If possible, the graphical user interface should be adapted to include some information on the possible signal that could be removed by applying the filter under consideration.

Minor remarks

1. Page 1. Title. I suggest to use 'Interactive' instead of the words 'human in the loop'. It is an elegant interactive approach, and I found the abbreviation HITL in the text not helping.
2. On several pages. I did not find the abbreviation w.r.t. (with respect to) useful.
3. Page 4. It is written at the bottom of the page 'state-of-the-art noise level estimator', and a reference to a proceedings paper from 2012 is used (Liu). As this estimator provides a key performance indicator of the noise reduction, it would be good to expand on this in the supplemental information.

Reviewer #2 (Remarks to the Author):

In their manuscript 'A "Human-in-the-Loop" Approach for Semi-automated Image Restoration in Electron Microscopy', the authors describe their software DenoisEM, for interactive image denoising.

DenoisEM is an ImageJ plugin that makes 8 established denoising methods for 2D images available to ImageJ users. The methods are implemented using the Quasar engine and can make use of the GPU if available on the system which makes them more efficient than existing CPU based implementations in ImageJ or Matlab.

The plugin enables users to explore and optimize parameters of the methods by generating interactive previews. It also has a 'wizard' that estimates good parameter choices for given input images.

I find the described software a useful and helpful contribution to the field of microscopy as it makes fast implementations of current denoising methods available. The interactive preview mode, similar to other existing filters in ImageJ, is helpful to explore appropriate parameter settings. The wizard helps users to seed their parameter search.

The parameters that are eventually applied to an image are stored as meta-data in custom tags using ImageJ's file format support (if the target file format supports this, i.e. TIFF).

The authors show that their implementation generates good denoising results fast and that the resulting denoised images can be analyzed easier with simple noise-sensitive methods such as naive thresholding or ilastik's random forest classifiers.

I find the manuscript, in its current form, unnecessarily difficult to read. My main issue is that it is not clear from the title, the abstract, and significant parts of the introduction that the authors describe a tool for image denoising. Instead, denoising filters with preview are re-coined as a supposedly new and revolutionary HITL approach to image restoration. This approach is not revolutionary. All existing tools, particularly filters in ImageJ are in one way or another HITL in that they generate previews to support manual parameter tuning.

The abstract and introduction motivate the work with 3D EM, but the proposed methods are all 2D. I find this confusing. The authors anonymously dismiss existing automatic analysis methods (vaguely throwing it all together) as insufficient and manual methods as biased. Their proposed solution is to manually tune denoising algorithms with interactive preview. I find this illogical and would prefer a more neutral description. The authors developed a great tool that does its job well and fast and is user-friendly. I would strongly prefer it described as such and not as a 'new paradigm' which it isn't.

Comments as they arose while reading:

List 3D EM methods or do not motivate with 3D EM.

strike [...]which has several issues[...]

segmentation is identified as the bottleneck, then thresholding is mentioned, then manual segmentation (but not what kind of manual segmentation).

[...]prompting the introduction of quality control. What does this mean?

How is a mathematical formulation less biased than human expertise? Isn't it biased towards the assumptions/ simplifications of the model?

"[...]users tend to be subjective and biased towards interactive image restoration in EM, suggesting that a 'human-in-the-loop' (HITL) can determine the optimal parameter settings for efficiency and reproducibility."

This means that "HITL transfers user bias to automatic pipelines" Is that what you want to say? If so, how is that a good thing?

Oh, their actual topic is denoising, not segmentation, revealed at the end of the introduction.

"Each step is automated as much as possible and user interaction is only required in the selection of the ROI and parameter settings." This is exactly what all other plugins with interactive preview in ImageJ are doing. The 'HITL approach' is a common sense standard method. I do not think that we need this word here. How about "plugin with interactive preview and parameter wizard"?

How are "near optimal" parameters estimated?

How can cached parameters or parameters stored in image meta-data be applied to other images (to reproduce results or to apply to similar data).

Figure 3 YZ plane non-isotropic and too small to see meaningful data. Also, since the methods work in 2D only, not necessary. Denoised images look blurry. How does Gaussian smoothing do in comparison?

Figure 4. I believe that simple Gaussian smoothing (suppression of high frequencies) would result in similar effects.

Figure 6. cremi.org has ground truth segmentation but no membrane ground truth. Where does the ground truth come from? How much data and which data has been used?

Reply to the referees' comments on manuscript "A Human-in-the-Loop Approach for Semi-automated Image Restoration in Electron Microscopy"

We would like to thank the reviewers and the editor for the opportunity to revise our manuscript. This document addresses the reviewers' concerns and explains the changes that were made to the first submission of "A Human-in-the-Loop Approach for Semi-automated Image Restoration in Electron Microscopy" in accordance with the comments. A revised version of the manuscript was submitted along with this document.

Reviewer 1

- *The paper describes a workflow (DenoisEM) to improve the signal to noise ratio of data sets acquired by serial block face scanning electron microscopy (SBF-SEM). The core of the workflow is an interactive graphical display that connects 'expert human eye' to the effectiveness of a set of effective noise-reducing algorithms (wavelet thresholding, anisotropic diffusion, bilateral filtering, Tikhonov denoising, total variation denoising, BLS-GSM, non-local means denoising, non-local means deconvolution). The effectiveness of the algorithms are evaluated on a several large SBF-SEM data sets. The DenoisEM workflow is a hybrid approach, bridging the application of noise-reducing algorithms to real-life datasets that required a human expert-eye to judge whether to noise-reduction is indeed carried out to a level that the biological question requires.*

A strong point of the manuscript is the notion that having in place a high-quality well-defined step of enhancing the signal to noise will have the effect that the data collection speed can be drastically reduced.

The implementation of DenoisEM is transparent and pragmatic, is a plugin for ImageJ and is tailed to include high performance computing using GPUs.

- *The manuscript is well written and the results are presented clearly. Nevertheless, the English and story flow can be improved at several instances.*

Authors' response: We thank the reviewer for this remark and have revised the complete manuscript to improve readability. We hope that the adjustments are sufficient for the reviewer.

- *I have a few general comments that would possible improve the impact the manuscript could have on the EM field.*
 1. *A strong point of the manuscript is the notion that by performing high quality noise filtering after data collection the actual data collection time could be improved (reduced) significantly, possibly an order of magnitude. This is important, as data collection with SBF-SEM can take hours and in most cases a day or longer.*

However, this argument is not mentioned in the abstract, nor highlighted in the introduction. It is mentioned in the last paragraph of the discussion.

In order to make this point convincingly experimental data is lacking. It would be convincing if the supplemental data could be extended by a data set collected with a long dwell time (high signal to noise) and of the same specimen block a data set collected with a short dwell time (low signal to noise), e.g. with a 5 or 10 times lower dwell time. If indeed the quality of the data sets (based upon visual inspection) after processing are very similar, the argument that the throughput of data collection time is increased can be made.

It would good if such data could be added to the supplement.

Authors' response: We agree with the reviewer that this point is an important one and should be emphasized. We have included the suggested experiment and validated that, for a dataset acquired at a 4 μ s dwell-time, similar image quality can be obtained by acquiring it at a 4 faster dwell-time and subsequent denoising.

- 2. *I miss a discussion in the manuscript on the interplay between noise filtering and resolution loss.*

In general, there is an inherent unwanted effect that by applying a filter not only (statistical) noise is reduced, but also a low-pass filtering effect is introduced that could reduce the resolution of the data set (blurring the edges). An effect that is clearly illustrated in supplementary figure 8, comparing inset (h) with inset (b).

Figure 1. Original and denoised images along with noise and blur metrics (top row) and their corresponding power spectrum (bottom row).

To help the user decide in choosing the filter parameters, I believe an addition to the graphical user interface is necessary that indicates the effect the filter will have on the resolution/sharpness of the data sets. If such an addition is not available, it will be likely that datasets are ‘over processed’, reducing more resolution/sharpness than necessary.

Perhaps that a pragmatic solution to providing such information would be to compute for a number of 2D SBF-SEM digital slices the power spectrum (e.g. for 10-50 slices), and average those power spectra. This would provide a good measure of the actual signal spectrum in the 2D slices of the data set. Next, one could do the same for the data sets that are processed by the noise removal filter that is used to process that data. As a measure one could provide a ‘number’ that indicates what percentage of the signal spectrum is retained after applying a noise removal filter. Ideally, the percentage would be ‘100%’, but one can imagine that in practice pragmatic - but reproducible - choices can be made.

The section in the manuscript on the discussion of possible loss of resolution/sharpness by the application of the noise reduction filters should be aimed at not ‘overprocessing’ data.

This section should also provide a means for the user on how to make sure that not too much signal is scarified relative to the amount of noise that is. If possible, the graphical user interface should be adapted to include some information on the possible signal that could be removed by applying the filter under consideration.

Authors’ response: We agree with the reviewer that denoising can introduce edge blurring, depending on the parameter settings, and should therefore be discussed in more detail. Therefore, we have added an additional note in the discussion of the revised manuscript.

We thank the reviewer for the interesting idea of adding a blur estimation method (*i.e.* high values for blurry data and vice versa) based on power spectrum (PS) overlap. However, it is important to keep in mind that the image noise also affects the PS and therefore the blur estimate. As an alternative, we have tested and implemented an intensity-based and noise robust blur estimator¹.

We have performed the experiment suggested by the reviewer on a dataset from the paper as well as on the denoised image (optimized by an expert), and several low-pass filtered versions (see Figure 1). Our conclusions are as follows:

- * The PS-based blur metric is always 100% for the noisy input image, even if it already is blurred (e.g. due to improper focusing). The proposed blur metric¹ does not have this bias.
- * Both the PS-based and proposed blur metric¹ are decreasing as the degree of denoising increases, which is according to our expectations.
- * The difference between the spectra of the input image and the restored versions is rather small, except for more aggressive denoising settings (e.g. Gaussian filtering with $\sigma = 5$).

We propose to include the proposed blur metric¹ in DenoisEM. However, in order to avoid users to maximize performance with respect to this measure, the blur metric will not be visible by default. We propose to visualize this measure upon request in the interface. Note that we have also included this in the description of the plugin in the manuscript. The top row of Figure 1 that illustrates the blur metric has also been added to the supplementary material.

- *Minor remarks*

1. Page 1. Title. I suggest to use 'Interactive' instead of the words 'human in the loop'. It is an elegant interactive approach, and I found the abbreviation HITL in the text not helping.

Authors' response: We agree with the reviewer and have replaced the 'HITL' terminology by 'interactive' across the complete manuscript.

- 2. On several pages. I did not find the abbreviation w.r.t. (with respect to) useful.

Authors' response: We agree with the reviewer and have removed this to increase the readability.

- 3. Page 4. It is written at the bottom of the page 'state-of-the-art noise level estimator', and a reference to a proceedings paper from 2012 is used (Liu). As this estimator provides a key performance indicator of the noise reduction, it would be good to expand on this in the supplemental information.

Authors' response: This was a typographical error in the text, as the current version of the plugin uses median absolute deviation for noise estimation. This method is also described in the methods section. We thank the reviewer for pointing out this error.

Reviewer 2

- In their manuscript 'A "Human-in-the-Loop" Approach for Semi-automated Image Restoration in Electron Microscopy', the authors describe their software DenoisEM, for interactive image denoising.

DenoisEM is an ImageJ plugin that makes 8 established denoising methods for 2D images available to ImageJ users. The methods are implemented using the Quasar engine and can make use of the GPU if available on the system which makes them more efficient than existing CPU based implementations in ImageJ or Matlab.

The plugin enables users to explore and optimize parameters of the methods by generating interactive previews. It also has a 'wizard' that estimates good parameter choices for given input images.

I find the described software a useful and helpful contribution to the field of microscopy as it makes fast implementations of current denoising methods available. The interactive preview mode, similar to other existing filters in ImageJ, is helpful to explore appropriate parameter settings. The wizard helps users to seed their parameter search.

The parameters that are eventually applied to an image are stored as meta-data in custom tags using ImageJ's file format support (if the target file format supports this, i.e. TIFF).

The authors show that their implementation generates good denoising results fast and that the resulting denoised images can be analyzed easier with simple noise-sensitive methods such as naive thresholding or ilastik's random forest classifiers.

- I find the manuscript, in its current form, unnecessarily difficult to read. My main issue is that it is not clear from the title, the abstract, and significant parts of the introduction that the authors describe a tool for image denoising. Instead, denoising filters with preview are re-coined as a supposedly new and revolutionary HITL approach to image restoration. This approach is not revolutionary. All existing tools, particularly filters in ImageJ are in one way or another HITL in that they generate previews to support manual parameter tuning.

Figure 2. Computational performance comparison of 2D and 3D based implementations.

Authors' response: We agree with the reviewer. A significant part of the abstract and introduction was focused on image segmentation to illustrate the importance of interactive semi-automated image processing. We have removed these parts of the text, the emphasis is now much more on the necessity for semi-automated image denoising in volume EM. More specifically, we state that the existing state-of-the-art denoising algorithms are impractical due to low-level programming environments, tedious parameter tuning or computational workload. Our plugin solves these issues through GPU acceleration and a practical user interface.

- *The abstract and introduction motivate the work with 3D EM, but the proposed methods are all 2D. I find this confusing.*

Authors' response: All methods are indeed implemented in 2D, 3D datasets are processed in a slice-by-slice fashion. Nevertheless, all of the discussed algorithms can be implemented in 3D, but this comes at the cost of memory and computation time. In fact, we have implemented the Gaussian filter, bilateral filter and tikhonov denoising in 2D and 3D and concluded that the compute time of the 3D variants are typically 50% longer (see Figure 2).

Below are several examples of 2D vs 3D denoising with the aforementioned techniques. As expected, the restoration quality has slightly increased (see Figure 3). However, we are not yet convinced whether this is worth the additional processing time. Additionally, the 3D implementations require much more memory, which is especially an issue for GPU-based processing. For example, most of the methods fail at data volumes larger than 0.5 gigavoxel.

Note that we have included this remark in the methods section at the beginning of the algorithm descriptions. We also provided the visual and computational comparison of the 2D and 3D based implementation in the supplementary material. We thank the reviewer for this comment and consider the full 3D extensions implementation in the plugin as future work.

- *The authors anonymously dismiss existing automatic analysis methods (vaguely throwing it all together) as insufficient and manual methods as biased. Their proposed solution is to manually tune denoising algorithms with interactive preview. I find this illogical and would prefer a more neutral description. The authors developed a great tool that does its job well and fast and is user-friendly. I would strongly prefer it described as such and not as a 'new paradigm' which it isn't.*

Authors' response: We understand the concerns of the reviewer and have rewritten the abstract and introduction thoroughly with a particular emphasis on the importance of image denoising, an interactive interface and computational efficiency. These are the main points, as the reviewer refers, of our manuscript.

- *Comments as they arose while reading:
List 3D EM methods or do not motivate with 3D EM.*

Authors' response: We have included references to 3D EM methods²⁻⁴ in the algorithm description part of the methods section. We additionally motivate the 2D slice-by-slice denoising as previously described.

- *strike [...]which has several issues[...]*

Authors' response: We thank the reviewer for this remark and have removed this part.

Figure 3. Denoising performance comparison of 2D and 3D based implementations. Each subfigure shows a cross-section along the z , y and x axis (from left to right, respectively).

- *segmentation is identified as the bottleneck, then thresholding is mentioned, then manual segmentation (but not what kind of manual segmentation).*

Authors' response: This is a valid point of the reviewer. In the initially submitted manuscript we always mean 'manual pixel delineation of object boundaries' by 'manual segmentation'. As the reviewer suggested, the manuscript should focus more towards denoising instead of segmentation. As such, there are no more references to manual segmentation in the new version.

- *...prompting the introduction of quality control. What does this mean?*

Authors' response: This means that the quality of the data should be checked and preprocessing should be considered. We understand the question of the reviewer and have rewritten this part of the text to improve readability.

- *How is a mathematical formulation less biased than human expertise? Isn't it biased towards the assumptions/simplifications of the model?*

Authors' response: We agree with the reviewer and find this an interesting question. We consider human bias to be more dangerous as this could potentially lead to data processing that benefits the final results. The reviewer is correct in the fact that there is indeed a mathematical bias towards the assumptions of the model, but these are usually designed to be as realistic as possible.

- “[...]users tend to be subjective and biased towards interactive image restoration in EM, suggesting that a ‘human-in-the-loop’ (HITL) can determine the optimal parameter settings for efficiency and reproducibility.”

This means that “HITL transfers user bias to automatic pipelines” Is that what you want to say? If so, how is that a good thing?

Oh, their actual topic is denoising, not segmentation, revealed at the end of the introduction.

Authors' response: We understand that the reviewer is confused in this part of the text. The focus is now more towards semi-automated denoising in the title, abstract and introduction. There are considerably less references to segmentation to avoid confusion.

- “Each step is automated as much as possible and user interaction is only required in the selection of the ROI and parameter settings.” This is exactly what all other plugins with interactive preview in ImageJ are doing. The ‘HITL approach’ is a common sense standard method. I do not think that we need this word here. How about “plugin with interactive preview and parameter wizard”?

Authors' response: We agree with the reviewer and have replaced the ‘HITL’ terminology by ‘interactive’ across the complete manuscript.

- How are “near optimal” parameters estimated?

Authors' response: To answer this question, we refer to the ‘Parameter estimation’ subsection of the methods section. Briefly speaking, we compiled a set of images, added different levels of noise to these images, applied denoising with various parameter settings and optimized with respect to peak signal to noise ratio. Finally, we performed linear regression to find the relation between the noise level and the optimal parameter settings. We have included figures of these regression models to the supplementary, the code to generate these results are in the supplementary code.

- How can cached parameters or parameters stored in image meta-data be applied to other images (to reproduce results or to apply to similar data).

Authors' response: We understand that this is unclear to the reviewer as this is insufficiently well explained in the manuscript. Loading parameters works as follows.

In the parameter tuning step, the user can load an image. If that image was generated by our plugin, it will contain metadata on the algorithm and parameters settings used for processing that particular dataset. These settings will then be forwarded to the plugin and applied on the reference data.

In the future, we will save the algorithm and parameters in a separate file that can then be loaded later for application on new data.

We thank the reviewer for this remark and have provided the text with additional information about this.

- Figure 3 YZ plane non-isotropic and too small to see meaningful data. Also, since the methods work in 2D only, not necessary. Denoised images look blurry. How does Gaussian smoothing do in comparison?

Authors' response: We have selected an isotropic YZ region and enlarged the image for better visualization. The orthogonal views and noise estimations show that, even though the methods are 2D, they also improve image quality in the third dimension. Additionally, experts have validated in these images that ultrastructure such as the nuclear membrane and endoplasmic reticulum is easier to recognize. For these reasons, we believe that the YZ and XZ planes are necessary.

Nevertheless, it is not the aim of this manuscript to compare different denoising algorithms. We believe there is no one-fits-all denoising algorithm that will perform optimally on each dataset. This is exactly the motivation to offer several well-performing methods in one accessible plugin.

If the reviewer is interested in a performance comparison of state-of-the-art denoising algorithms, we would like to refer to our previous work⁵.

- *Figure 4. I believe that simple Gaussian smoothing (suppression of high frequencies) would result in similar effects.*

Authors' response: We agree with the reviewer. In fact, during the denoising of this particular dataset, experts of our group also noted that the quality difference with Gaussian smoothing was subtle. We recall that it is not our purpose to compare the performance of different denoising algorithms in this manuscript.

- *Figure 6. cremi.org has ground truth segmentation but no membrane ground truth. Where does the ground truth come from? How much data and which data has been used?*

Authors' response: The CREMI dataset contains annotations of the different neurons. We extracted the outer neuron membranes by computing the edges between the different neurons (using a Sobel filter) and dilating these edges by 5 pixels, the mean membrane width in the dataset.

We thank the reviewer for this remark and have added this to the text.

References

1. Crete, F., Dolmiere, T., Ladret, P. & Nicolas, M. The blur effect: perception and estimation with a new no-reference perceptual blur metric. In *Human Vision and Electronic Imaging XII* (2007). DOI 10.1117/12.702790.
2. Manjón, J. V., Coupé, P., Buades, A., Louis Collins, D. & Robles, M. New methods for MRI denoising based on sparseness and self-similarity. *Med. Image Analysis* (2012). DOI 10.1016/j.media.2011.04.003.
3. Maggioni, M., Katkovnik, V., Egiazarian, K. & Foi, A. Nonlocal transform-domain filter for volumetric data denoising and reconstruction. *IEEE Transactions on Image Process.* **22**, 119–133 (2013). DOI 10.1109/TIP.2012.2210725.
4. Cuomo, S., De Michele, P. & Piccialli, F. 3D data denoising via Nonlocal means filter by using parallel GPU strategies. *Comput. Math. Methods Medicine* **2014** (2014). DOI 10.1155/2014/523862.
5. Roels, J. *et al.* An Overview of State-of-the-art Image Restoration in Electron Microscopy. *J. Microsc.* (2018).

REVIEWERS' COMMENTS:

Reviewer #1 (Remarks to the Author):

I have evaluated the revised manuscript and am satisfied that the points raised in the previous round of review were satisfactorily addressed.

Reviewer #2 (Remarks to the Author):

This version is much easier to read and understand, thank you!

Comments:

Introduction

Zheng et al. 2018 is not SEM but a TEM series and therefore not a good example for the section. This is particularly confusing when, at page 7, the authors describe the CREMI challenge dataset as approximately noise-free). The CREMI challenge data is part of the Zheng et al. 2018 volume.

Figure 1 caption: How is the "noise level [...] automatically estimated to derive near optimal parameter initialization"? Please reference the parameters section and the supplement.

How can the parameters that are stored in TIFF tags be used to denoise a new image? (I understand that you answered that in the response but it's not in the text, please reference supplement here).

Is the plugin macro-friendly?

page 5 EM imaging throughput

The paragraph is missing a discussion of the resolution vs denoising. Another trivial way to reduce the imaging time by a factor of 4 would be to image at 4us dwell time with 20nm pixels. Denoising would be a superior solution only if high-resolution features were preserved. Figure 5 does not support this and I suggest to perform a quantitative analysis (look at the frequency spectrum).

page 10

"For example, expert CUDA developers required three months to implement an MRI algorithm[44], whereas a single Quasar developer achieved the same numerical results at the same computational performance within a week"

I am not impressed by this sentence and suggest to cite the Quasar paper instead and say something more general about the benefits of the high-level API (e.g. that it does not only run on Nvidia hardware such as CUDA?). The sentence in its current form does not describe the programming task nor the algorithm nor the context and is therefore not very helpful. Why not write something like

"The high-level Quasar API has been demonstrated to support rapid software development without compromising performance in direct comparison with other popular GPU development platforms such as Cuda (see [44] and Cuda)."

page 11

"We show on data that we can increase the throughput of 3D EM by a factor of 4 without affecting image quality."

See my comments above and look at resolution, also strike "on data".

page 12

"Consequently, the acquisition time can be shortened to increase throughput or avoid overexposure, without significantly affecting the subsequent segmentation."

This is a bit strong and generic. This paper demonstrates that denoising is helpful for simple thresholding based analysis pipelines and a shallow pixel classifier. State of the art classifiers have not been investigated and the results may be different (deep learning based classifiers, e.g., typically learn how to denoise and have been shown to do this well, but this is not the topic of this paper). Instead, you demonstrated that for some simple segmentation or classification tasks, denoised low SNR images are not worse than high SNR images and this can be used in practice to reduce imaging time.

Supplement and user manual very nice.

Reply to the referees' comments on manuscript "DenoisEM: An Interactive ImageJ Plugin for Semi-automated Image Denoising in Electron Microscopy"

We would like to thank the reviewers and the editor for the opportunity to revise our manuscript. This document addresses the reviewers' concerns and explains the changes that were made to the second submission of "DenoisEM: An Interactive ImageJ Plugin for Semi-automated Image Denoising in Electron Microscopy" in accordance with the comments. A revised version of the manuscript was submitted along with this document.

Reviewer 1

- *I have evaluated the revised manuscript and am satisfied that the points raised in the previous round of review were satisfactorily addressed.*

Reviewer 2

- *This version is much easier to read and understand, thank you!*
- *Introduction*

Zheng et al. 2018 is not SEM but a TEM series and therefore not a good example for the section. This is particularly confusing when, at page 7, the authors describe the CREMI challenge dataset as approximately noise-free). The CREMI challenge data is part of the Zheng et al. 2018 volume.

Authors' response: We agree with the reviewer and have replaced the example of Zheng et al. 2018 by that of Xu et al. 2018 and Xu et al. 2019, two papers on high throughput FIB-SEM imaging.

We understand the second remark, regarding the CREMI dataset. Nevertheless, we would also like to clarify that DenoisEM can be used for any type of 3D dataset, including serial section TEM. For this reason, we have chosen to leave the CREMI example in the results. The introduction now also contains a reference to serial section TEM and the first paragraph in the results section emphasizes that our plugin can serve for both SEM and TEM, or in fact any modality that generates 3D image data.

Figure 1 caption: How is the "noise level [...] automatically estimated to derive near optimal parameter initialization"? Please reference the parameters section and the supplement.

Authors' response: We have added the requested references.

- *How can the parameters that are stored in TIFF tags be used to denoise a new image? (I understand that you answered that in the response but it's not in the text, please reference supplement here).*

Authors' response: We agree with the reviewer and have provided this information in the first part of the results section (last bullet point).

- *Is the plugin macro-friendly?*

Authors' response: Currently, the plugin does not support macro scripting, but we are considering this for future work.

- *page 5 EM imaging throughput*

The paragraph is missing a discussion of the resolution vs denoising. Another trivial way to reduce the imaging time by a factor of 4 would be to image at 4us dwell time with 20nm pixels. Denoising would be a superior solution only if high-resolution features were preserved. Figure 5 does not support this and I suggest to perform a quantitative analysis (look at the frequency spectrum).

Figure 1. (a) SBF-SEM section acquired at 1 μ s dwell-time, 8 nm pixel size and denoised by an expert using DenoisEM, (d) the same image, acquired at 4 μ s dwell-time, 16 nm pixel size, which results in approximately the same acquisition time. The corresponding Fourier spectra (b,e) show that our the denoising algorithms do not significantly affect the high-frequency components of the image. When masking out the low-frequency components (indicated by the yellow circle), we can reconstruct the image and see the resolution improvements in the spatial domain (c,f).

Authors' response: The reviewer is correct that increasing the pixel size is an alternative method to accelerate imaging and we should include a discussion on resolution in the manuscript. Therefore, we have repeated the complete experiment, described in this section. We have now also included data acquired at twice the pixel size (see figure 1). Next, we have analyzed the Fourier spectra and validated that our denoised results removes noise without significantly affecting the high-frequency (HF) components. By masking out the low-frequency components, we can also visualize this in the image domain by applying the inverse Fourier transform. Clearly, the image acquired at a larger pixel size is less detailed and noisy. This discussion is now also included in the main paper and the figure is included in Supplementary material. We thank the reviewer for this suggestion.

- page 10

“For example, expert CUDA developers required three months to implement an MRI algorithm[44], whereas a single Quasar developer achieved the same numerical results at the same computational performance within a week”

I am not impressed by this sentence and suggest to cite the Quasar paper instead and say something more general about the benefits of the high-level API (e.g. that it does not only run on Nvidia hardware such as CUDA?). The sentence in its current form does not describe the programming task nor the algorithm nor the contest and is therefore not very helpful. Why not write something like

“The high-level Quasar API has been demonstrated to support rapid software development without compromising performance in direct comparison with other popular GPU development platforms such as Cuda (see [44] and Cuda).”

Authors' response: We agree with the reviewer and have replaced this by the suggested sentence.

- *page 11*

*“We show on data that we can increase the throughput of 3D EM by a factor of 4 without affecting image quality.”
See my comments above and look at resolution, also strike “on data”.*

Authors' response: We have removed “on data” in this sentence.

- *page 12*

“Consequently, the acquisition time can be shortened to increase throughput or avoid overexposure, without significantly affecting the subsequent segmentation.”

This is a bit strong and generic. This paper demonstrates that denoising is helpful for simple thresholding based analysis pipelines and a shallow pixel classifier. State of the art classifiers have not been investigated and the results may be different (deep learning based classifiers, e.g., typically learn how to denoise and have been shown to do this well, but this is not the topic of this paper). Instead, you demonstrated that for some simple segmentation or classification tasks, denoised low SNR images are not worse than high SNR images and this can be used in practice to reduce imaging time.

Authors' response: We agree with the reviewer and have rewritten this sentence with a clear emphasis on thresholding and pixel classification.

- *Supplement and user manual very nice.*